# CLK2 mediates IκBα-independent early termination of NF-κB activation by inducing cytoplasmic redistribution and degradation

Shang-Ze Li [1,2,5], Qi-Peng Shu [1,5], Hai-Meng Zhou[1], Yu-Ying Liu [1], Meng-Qi Fan [1], Xin-Yi Liang [1], Lin-Zhi Qi [2], Ya-Nan He [1], Xue-Yi Liu [1], Xue-Hua Du [1], Xi-Chen Huang [1], Yu-Zhen Chen [3], Run-Lei Du [1]✉, Yue-Xiu Liang [3]✉ & Xiao-Dong Zhang [1,3,4]✉

Activation of the NF-κB pathway is strictly regulated to prevent excessive inflammatory and immune responses. In a well-known negative feedback model, IκBα-dependent NF-κB termination is a delayed response pattern in the later stage of activation, and the mechanisms mediating the rapid termination of active NF-κB remain unclear. Here, we showed IκBα-independent rapid termination of nuclear NF-κB mediated by CLK2, which negatively regulated active NF-κB by phosphorylating the RelA/p65 subunit of NF-κB at Ser180 in the nucleus to limit its transcriptional activation through degradation and nuclear export. Depletion of CLK2 increased the production of inflammatory cytokines, reduced viral replication and increased the survival of the mice. Mechanistically, CLK2 phosphorylated RelA/p65 at Ser180 in the nucleus, leading to ubiquitin–proteasome-mediated degradation and cytoplasmic redistribution. Importantly, a CLK2 inhibitor promoted cytokine production, reduced viral replication, and accelerated murine psoriasis. This study revealed an IκBα-independent mechanism of early-stage termination of NF-κB in which phosphorylated Ser180 RelA/p65 turned off posttranslational modifications associated with transcriptional activation, ultimately resulting in the degradation and nuclear export of RelA/p65 to inhibit excessive inflammatory activation. Our findings showed that the phosphorylation of RelA/p65 at Ser180 in the nucleus inhibits early-stage NF-κB activation, thereby mediating the negative regulation of NF-κB.

Inflammation and innate immunity are self-defense mechanisms used by the host to combat pathogens[1,2]. These processes are involved in various physiological and pathological processes and are regulated by multiple signaling pathways[3]. Nuclear factor kappa B (NF-κB), which is a key molecule in inflammation and innate immunity[4–6], is associated with defense against pathogens and various inflammatory diseases[7,8]. A lack of p65, which is an NF-κB subunit, significantly reduces the production of inflammatory cytokines, thereby increasing host vulnerability to viral infection[9,10] and highlighting the importance of NF-κB in protecting against pathogen invasion[5,11–13].

[1]Hubei Key Laboratory of Cell Homeostasis, College of Life Sciences, Wuhan University, Wuhan 430072 Hubei, China. [2]School of Medicine, Chongqing University, Chongqing 400044, China. [3]Key Laboratory of Research on Clinical Molecular Diagnosis for High Incidence Diseases in Western Guangxi of Guangxi Higher Education Institutions & Department of Gynecology, Affiliated Hospital of Youjiang Medical University for Nationalities, Baise, China. [4]National Health Commission Key Laboratory of Birth Defect Research and Prevention & MOE Key Lab of Rare Pediatric Diseases, Department of Cell Biology and Genetics, School of Basic Medical Sciences, Hengyang Medical School, University of South China, Hengyang, China. [5]These authors contributed equally: Shang-Ze Li, Qi-Peng Shu. ✉e-mail: runleidu@whu.edu.cn; 105706981@qq.com; zhangxd@whu.edu.cn

The NF-κB transcription factor family is composed of five members (p105/p50, p100/p52, RelA/p65, RelB and c-Rel), which all have a Rel homology domain (RHD) at the N-terminus. The RHD contains a nuclear localization sequence (NLS) and is responsible for dimerization and binding with inhibitory κB proteins (IκBs)[14–16]. In response to specific stimuli, NF-κB proteins form various homodimers and heterodimers based on the RHD. Of these, the p50/p65 heterodimer is the most common and has been studied the most extensively. When activated, p50/p65 translocates into the nucleus[17], where it induces the production of inflammatory cytokines, such as interleukins and interferon, to eliminate pathogens.

Abnormal activation of NF-κB has been associated with many inflammation-related diseases, such as rheumatoid arthritis, psoriasis and HIV-1 infection[18–22]. Therefore, the activation of NF-κB is tightly regulated at different times and in different sites; for example, p65 is located in the cytoplasm via the binding of IκBα to p50/p65, the transcriptional activity of p65 is modified by posttranslational modifications, and IκBα restricts p65[23,24]. The traditional IκBα-dependent negative feedback loop involves the reassembly of the nuclear NF-κB dimer with IκBα and its transport out of the nucleus[25–28]. This feedback loop, however, ignores IκBα-independent, nuclear, and transient inhibitory transcriptional responses. The inhibition of NF-κB activation during the period between IκBα degradation and the re-synthesis of IκBα, which is a rapid IκBα-independent termination mechanism, has rarely been reported. Ubiquitination and deacetylation of p65 can terminate activation at the early stage[29–31]. How activated nuclear forms of NF-κB are negatively regulated by phosphorylation and whether these modifications are regulated are still largely unknown. In recent decades, significant progress has been made in understanding nuclear import and transcriptional activation of p65; for example, the phosphorylation of Ser536 and Ser276 is essential for transactivation, the phosphorylation of Ser468 may negatively regulate NF-κB-mediated transcription, and the acetylation of Lys310 is required for the full transcriptional activity of p65 subsequent to the phosphorylation of p65 at Ser276 and Ser536[29,32–35]. However, the mechanism by which the kinase responsible for phosphorylating active nuclear NF-κB leads to the termination and nuclear export of p65 remain poorly defined.

In this study, we show that IκBα-independent nuclear NF-κB termination occurs at the transcriptional level through inhibitory phosphorylation by CDC-like kinase 2 (CLK2). The phosphorylation of p65 at Ser180 eliminates active posttranslational modifications (PTMs), resulting in the degradation and nuclear export of p65. Our findings demonstrate that knockout of CLK2 in human and mouse cells increases the induction of inflammatory and antiviral genes and impairs viral replication and viral infection. Compared with their wild-type littermates, Clk2-deficient mice had increased serum levels of inflammatory cytokines after viral infection and showed increased resistance to virus-induced death. CLK2 interacted with and phosphorylated p65 at Ser180 after p65 was imported into the nucleus, which led to the degradation of nuclear p65 and its subsequent export from the nucleus. Our findings revealed a phosphorylation site of p65 and highlighted the critical role of CLK2 in maintaining inflammatory and innate immune homeostasis. This finding has enhanced our understanding of the mechanism by which nuclear p65 is inactivated, which occurs rapidly in the early stage of NF-κB activation and coordinates with the IκBα-dependent negative feedback loop to form a complete negative regulatory system for p65.

## Results

### CLK2 is a negative regulator of NF-κB transcriptional activation
To characterize the kinase that maintains inflammation and innate immune homeostasis associated with virus-induced signaling, we screened a kinase library using an IFN-β luciferase reporter after SeV stimulation[36]. Previously, we reported that NLK phosphorylated MAVS

to mediate degradation and regulate the antiviral innate immune response[37]. Extended screening revealed that TAOK3, BLK, MAPK27 and HIPK1 could effectively inhibit SeV-induced IFN-β activation. Notably, CLK2 completely blocked SeV-induced IFN-β activation, indicating its important role in innate immunity (Fig. 1A). Additionally, CLK2 had little effect on IFN-γ-induced IRF1 activation, suggesting its specificity for the type I interferon response (Supplementary Fig. 1a). To evaluate the inhibitory effect of CLK2 on SeV-induced IFN-β luciferase reporter activation, a dose-response experiment was performed, and CLK2 suppressed SeV-induced IFN-β luciferase reporter activation (Fig. 1B). Real-time PCR analysis further confirmed that CLK2 inhibited SeV-induced transcriptional activation of IFN-β and RANTES (Fig. 1C). IFN-β activation required the coordination of NF-κB and IRF3[38,39]; accordingly, an interferon-stimulated response element (ISRE) that only required IRF3 activation and an NF-κB luciferase reporter was used to determine whether the CLK2-dependent inhibition of type I interferon was dependent on its inhibitory effect on ISRE or NF-κB signaling. As shown in Fig. 1D, E, SeV-induced NF-κB activation but not ISRE activation was inhibited by CLK2 in a dose-dependent manner, suggesting that CLK2 inhibited IFN-β activation by blocking NF-κB signaling. Given that NF-κB signaling is essential for inflammation and innate immunity, we further determined whether CLK2 could inhibit inflammatory cytokine-induced NF-κB activation. Luciferase assays were performed to assess TNF-α- and IL-1β-induced NF-κB activation, and the results showed that CLK2 inhibited NF-κB activation in a dose-dependent manner (Fig. 1F, G), suggesting that CLK2 may inhibit NF-κB activation induced by universal stimuli.

To determine whether CLK2 can be induced by viruses or inflammatory stimuli, we first analyzed the CLK2 promoter and identified two NF-κB binding motifs located 693 and 1314 base pairs upstream of the ATG site (Supplementary Fig. 1b). Subsequent examination of the mRNA and protein levels of CLK2 in response to SeV and TNF-α revealed dynamic changes in CLK2 protein expression compared to mRNA expression (Supplementary Fig. 1c and Fig. 1H). To further verify the involvement of p65 in the regulation of CLK2, p65-deficient HEK293T cells were used. The results showed that the expression of IFNB1 and TNF-α dramatically increased after stimulation in wild-type HEK293T cells but was inhibited in p65-deficient HEK293T cells, and there was only a slight increase in CLK2 mRNA (Supplementary Fig. 1d). In conclusion, these findings suggest that CLK2 is mainly regulated at the posttranscriptional level rather than at the transcriptional level and is a negative regulator of NF-κB signaling.

### CLK2 deficiency enhances the NF-κB-mediated inflammatory response
To investigate the role of CLK2 in the inflammatory response in vivo, we generated Clk2-deficient mice (Clk2−/−) via CRISPR/Cas9-mediated gene knockout. Sequencing and genotyping indicated that exons 1-5 were completely deleted in the edited allele, which resulted in inactivation of the Clk2 gene (Fig. 2A, B). There were no significant differences in body weight or immune cell populations in the thymus or spleen between Clk2−/− mice and their wild-type littermates (Supplementary Fig. 2a–c), suggesting that Clk2 is dispensable for lymphocyte development. Subsequently, bone marrow dendritic cells (BMDCs) and mouse lung fibroblasts (MLFs) were isolated from Clk2−/− mice and their wild-type littermates, and Clk2 deficiency in these cells was verified by real-time PCR (Fig. 2B). To evaluate the effect of Clk2 deficiency on inflammation, the cells were exposed to lipopolysaccharide (LPS), and Clk2−/− cells exhibited significantly increased expression of NF-κB-induced downstream genes compared to Clk2+/+ cells (Fig. 2C, D). Similar experiments were performed and revealed that Clk2 deficiency promoted the production of proinflammatory cytokines in response to TNF-α and IL-1β (Supplementary Fig. 2d, e).

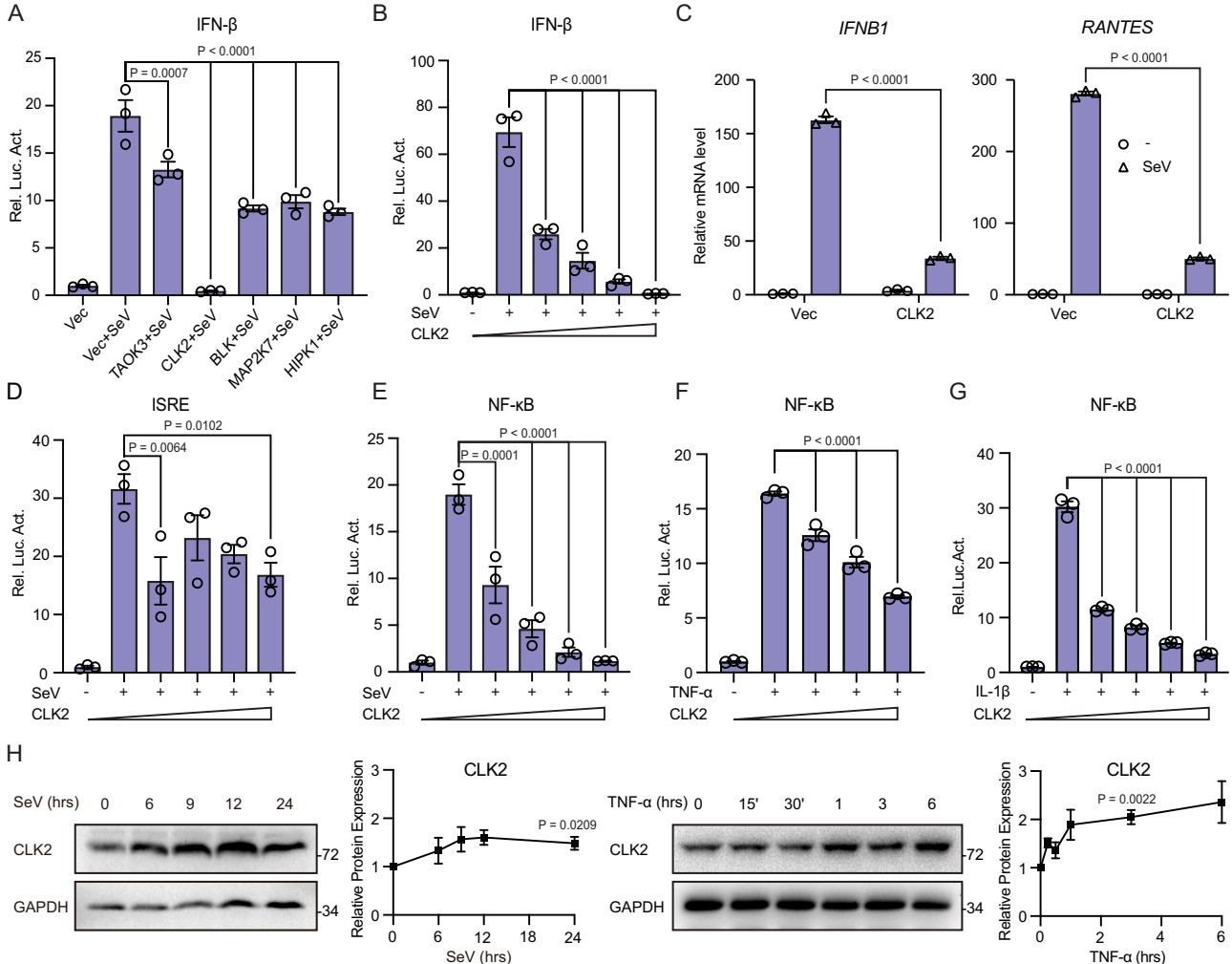

**Fig. 1 | CLK2 negatively regulates the transcriptional activation of NF-κB. A** The inhibitory effects of TAOK3, CLK2, BLK, MAP2K7 and HIPK1 on SeV-induced IFN-β promoter activity were determined by luciferase reporter assays in HEK293T cells cotransfected with an empty vector or plasmids encoding the indicated genes for 24 h followed by infection with SeV for 12 h. **B** The dose-dependent inhibitory effects of CLK2 on SeV-induced IFN-β promoter activity were determined by luciferase reporter assays in HEK293T cells transfected with increasing amounts of the CLK2 expression plasmid for 24 h followed by infection with SeV for 12 h. **C** The inhibitory effects of CLK2 on SeV-induced endogenous *IFNB1* and *RANTES* expression in HEK293T cells transfected with the Flag-CLK2 plasmid for 24 h followed by infection with SeV for 24 h were examined by real-time PCR assays. **D, E** The dose-dependent inhibitory effects of CLK2 on SeV-induced ISRE and NF-κB promoter activity in HEK293T cells transfected with increasing amounts of the CLK2 expression plasmid for 24 h followed by infection with SeV for 12 h were determined by luciferase reporter assays. **F, G** The dose-dependent inhibitory effects of CLK2 on TNF-α- or IL-1β-induced NF-κB promoter activity in HEK293T cells transfected with increasing amounts of the CLK2 expression plasmid for 24 h followed by infection with TNF-α or IL-1β for 6 h were determined by luciferase reporter assays. **H** The protein level of CLK2 in HEK293T cells after SeV or TNF-α stimulation at the indicated times were determined by Western blot analysis. The data are representative of three independent experiments. The data are presented as the means ± SEMs ($n = 3$ for **A**–**G**). Statistical significance was analyzed by two-tailed Student's *t* test (**A, B** and **D**–**H**) and two-tailed ANOVA (**C**) (*$p < 0.05$, **$p < 0.01$, ****$p < 0.0001$). Source data (**A**–**H**) are provided as a Source Data file.

The phosphorylation and acetylation of p65 are essential for DNA binding and transcriptional activation[29,32,40,41]. To evaluate whether Clk2 deficiency increased the activation of NF-κB signaling, the phosphorylation and acetylation of p65, as well as the level of Nfkbia (known as IκBα in humans), were measured by Western blotting after exposure to LPS, TNF-α, or IL-1β. As shown in Fig. 2E and Supplementary Fig. 2f, Clk2 deficiency increased the acetylation and phosphorylation of p65 during the later stage of stimulation. This finding suggested that CLK2 was involved in the activation of p65. Interestingly, there was an increase in Nfkbia accumulation in Clk2−/− cells, which was consistent with the increase in Lys310 acetylation and Ser536 phosphorylation of p65. This indicates that CLK2 may act as a basal checkpoint to limit the constitutive activation of NF-κB signaling, even in the absence of activating stimuli. Overall, CLK2 inhibits the NF-κB-mediated inflammatory response in primary cells derived from mice.

## CLK2 deficiency enhances virus-induced IFN-β production and the antiviral response

Because NF-κB is essential for the production of type I interferon, a series of assays were performed to investigate the effect of Clk2 deficiency on the virus-induced expression of downstream genes in BMDMs (Supplementary Fig. 3a), BMDCs, and MLFs. The results showed that the expression of Ifnb1 and Rantes was significantly increased in *Clk2−/−* cells in response to SeV or HSV-1 infection or after the transfection of various DNA ligands (Fig. 3A, B and Supplementary Fig. 3b–d). Furthermore, the replication of VSV-GFP in *Clk2−/−* cells was reduced compared to that in *Clk2+/+* cells, as determined by measuring the GFP intensity (Fig. 3C and Supplementary Fig. 3e). A plaque assay indicated that the level of VSV-GFP was reduced in *Clk2−/−* cells (Fig. 3D). In conclusion, Clk2 deficiency increased the production of downstream genes induced by different viruses and suppressed the replication of viruses in mouse cells.

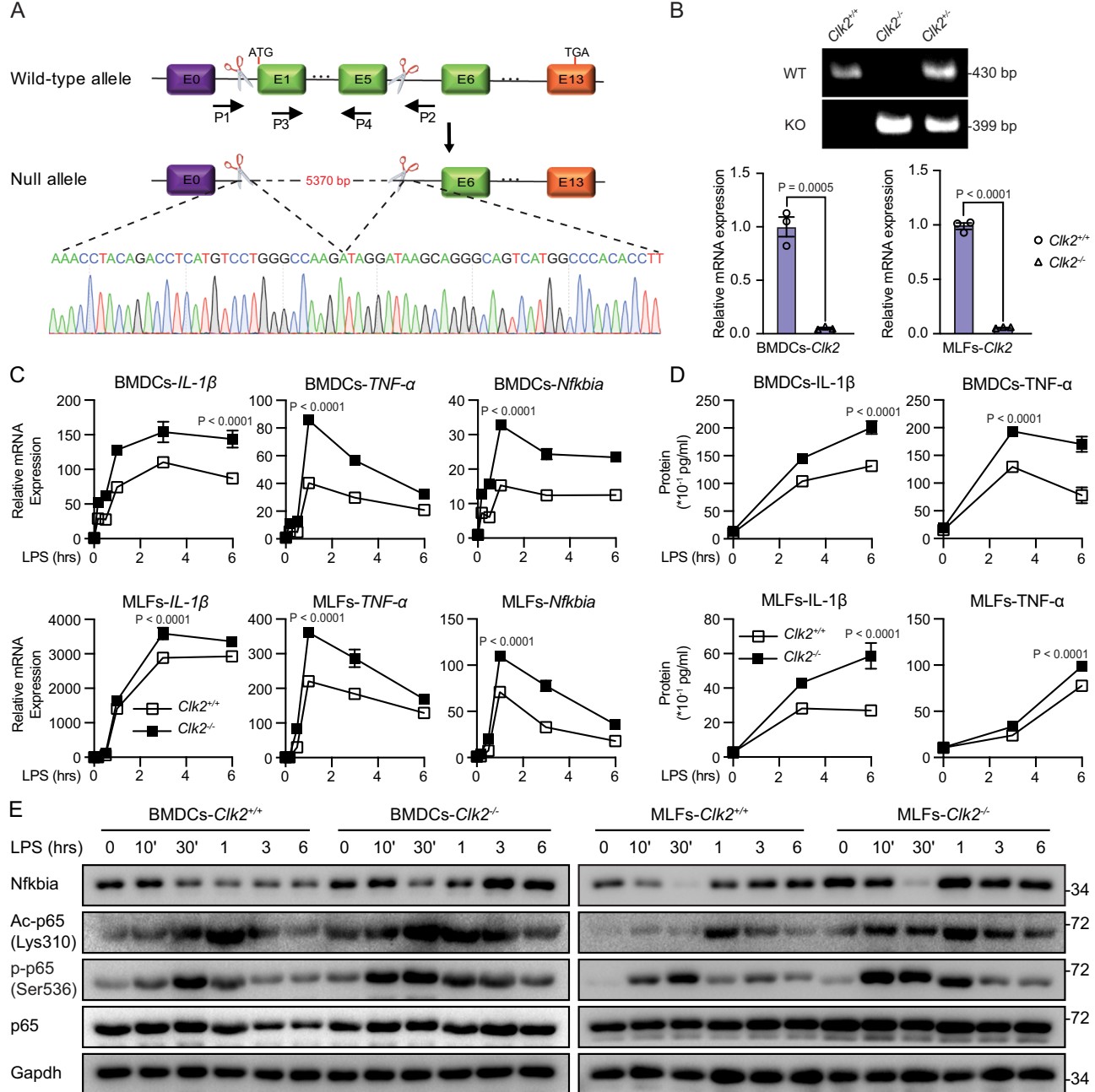

**Fig. 2 | Clk2 deficiency enhances the NF-κB-mediated inflammatory response. A** The targeting strategy for the deletion of exons 1–5 of *Clk2* and the DNA sequence of the *Clk2⁻/⁻* genome. **B** Generation and identification of Clk2-deficient mice at the genomic and mRNA levels (*n* = 3). The PCR products of Clk2^WT and Clk2^KO. **C** Real-time PCR analysis of *IL-1β*, *TNF-α* and *Nfkbia* mRNA levels in *Clk2⁺/⁺* and *Clk2⁻/⁻* BMDCs and MLFs stimulated with LPS (200 ng/ml) for the indicated times (*n* = 3). **D** ELISA analysis of IL-1β and TNF-α protein levels in *Clk2⁺/⁺* and *Clk2⁻/⁻* BMDCs and MLFs stimulated with LPS for the indicated times (*n* = 3). **E** Western blot analysis of Nfkbia, Ac-p65^K310, p-p65^S536, p65 and Gapdh in *Clk2⁺/⁺* and *Clk2⁻/⁻* BMDCs and MLFs stimulated with LPS for the indicated times. The data are representative of three independent experiments. The data are presented as the means ± SEMs (*n* = 3 for **B**–**D**). Statistical significance was analyzed by two-tailed Student's *t* test (**B**) or two-tailed ANOVA (**C**, **D**) (***p < 0.001, ****p < 0.0001). Source data (**A**–**E**) are provided as a Source Data file.

To assess the effect of CLK2 deficiency on type I interferon activation in human cells, CLK2⁻/⁻ HEK293T (human embryonic kidney cells) cells were generated using CRISPR/Cas9-mediated gene knockout (Supplementary Fig. 3g). We evaluated IRF3 phosphorylation during SeV-induced ISRE activation and found no major difference in IRF3 phosphorylation in response to SeV stimulation in CLK2-deficient cells, indicating that CLK2 inhibited transcriptional activation through the NF-κB cascade (Supplementary Fig. 3h). CLK2 deficiency significantly increased SeV-induced IFN-β and RANTES mRNA and protein production (Supplementary Fig. 3i, j). A viral replication assay revealed that CLK2-deficient

cells were more resistant to VSV-GFP replication than wild-type cells (Supplementary Fig. 3k, l). Collectively, these results suggest that CLK2 inhibits virus-induced signaling in mouse and human cells.

To investigate the physiological role of Clk2 in viral infection in vivo, Clk2-deficient mice and their wild-type littermates were injected with vesicular stomatitis virus (VSV) or herpes simplex virus type 1 (HSV-1) intraperitoneally and via the tail vein. We monitored the survival rates of the mice for 1 or 2 weeks and found that while the majority of wild-type mice died after viral injection, Clk2-deficient mice were resistant to lethal VSV or HSV-1 injection (Fig. 3E and

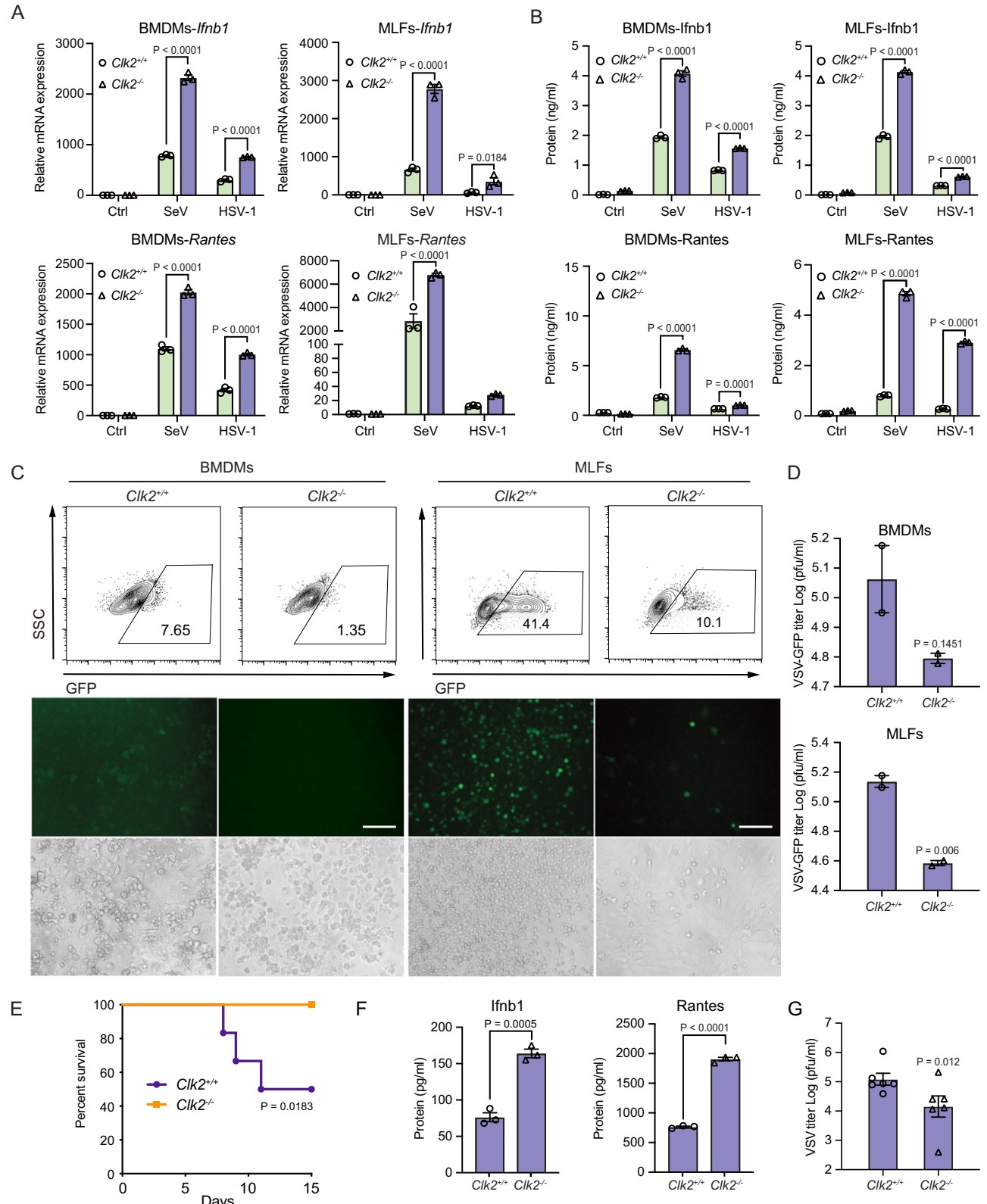

Supplementary Fig. 3f). Furthermore, the protein levels of Ifnb1 and Rantes in the sera of Clk2-deficient mice were notably higher than those in the sera of their wild-type littermates (Fig. 3F). Additionally, the viral titers in the sera of the $Clk2^{-/-}$ mice were lower than those in the sera of their wild-type littermates (Fig. 3G). Taken together, these results suggest that Clk2 deficiency significantly enhances the type I interferon response and antiviral immunity in vivo.

## CLK2 interacts with and phosphorylates p65 to reverse its nuclear localization

To determine the mechanism by which CLK2 regulates inflammatory and innate immune responses, the level of CLK2, which is involved in NF-κB signaling, was examined by luciferase reporter assays. CLK2 significantly inhibited luciferase reporter activation in the presence of RIG-I, MAVS, TAK1/TAB1, IKKβ, and p65, indicating that

**Fig. 3 | CLK2 deficiency enhances virus-induced IFN-β production and the antiviral response. A** Real-time PCR analysis of *Ifnb1* and *Rantes* mRN levels in *Clk2*⁺/⁺ and *Clk2*⁻/⁻ BMDMs and MLFs infected with SeV for 12 h or HSV-1 for 6 h (*n* = 3). **B** ELISA analysis of Ifnb1 and Rantes protein levels in *Clk2*⁺/⁺ and *Clk2*⁻/⁻ BMDMs and MLFs infected with SeV for 12 h or HSV-1 for 6 h (*n* = 3). **C** *Clk2*⁺/⁺ and *Clk2*⁻/⁻ BMDMs and MLFs were infected with VSV-GFP at an MOI of 2 for 12 h before phase contrast and fluorescence microscopy and flow cytometric analysis. The scale bar is 250 μm. **D** *Clk2*⁺/⁺ and *Clk2*⁻/⁻ BMDMs and MLFs were infected with VSV-GFP at an MOI of 2 for 1 h, followed by culture for 24 h with fresh medium. Then, the supernatants were diluted and added to Vero cells for plaque assays (*n* = 2).

**E** Survival assays (Kaplan–Meier curves) of wild-type and *Clk2*⁻/⁻ female mice at 6 weeks after infection with wild-type VSV (2 × 10⁷ pfu/g) via the tail vein and intraperitoneal injection. Mouse survival was monitored for 15 days (*n* = 6). **F** ELISA analysis of Ifnb1 and Rantes protein levels in the sera of VSV-infected *Clk2*⁺/⁺ and *Clk2*⁻/⁻ female mice at 24 h (*n* = 6). **G** Plaque assays in VSV-infected *Clk2*⁺/⁺ and *Clk2*⁻/⁻ mouse sera (*n* = 6). The data are representative of three independent experiments. The data are presented as the means ± SEMs. Statistical significance was analyzed by two-tailed Student's *t* test (**D**, **F**, **G**) or two-tailed ANOVA (**A**, **B**) (*p* > 0.05, not significant, **p* < 0.05, ****p* < 0.001, *****p* < 0.0001). Source data (**A**–**G**) are provided as a Source Data file.

CLK2 suppressed virus-induced NF-κB signaling by affecting p65 (Fig. 4A). Since the location of a protein is often closely associated with its function, the immunofluorescence assay further revealed that most CLK2 was located in the nucleus (Supplementary Fig. 4a), suggesting that CLK2 may affect nuclear p65 to regulate the activation of NF-κB signaling. To verify the association between CLK2 and p65, exogenous coimmunoprecipitation analyses were performed and showed that p65 interacted with CLK2 but not with CLK2^K192R (CLK2 kinase-dead mutant)[42] (Fig. 4B, C). Moreover, the ATP-competitive CLK2 inhibitor TG003 inhibited the interaction between CLK2 and p65 (Fig. 4D)[43], suggesting that the ability of CLK2 to bind to p65 depended on its kinase activity. To determine the efficiency of CLK2 antibody immunoprecipitation, Flag-tagged CLK2 knock-in HEK293T cells (CLK2^Flag/KI) were generated. Endogenous coimmunoprecipitation performed with a Flag antibody revealed that CLK2 weakly interacted with p65 in unstimulated cells, and the interaction was significantly strengthened after stimulation, suggesting that CLK2 interacts with p65 in the presence of virus or TNF-α (Fig. 4E). To further confirm whether the participation of CLK2 in NF-κB signaling is dependent on its kinase activity, we performed luciferase analyses, which showed that CLK2^K192R completely rescued the inhibition of TNF-α- and SeV-induced NF-κB activation (Supplementary Fig. 4b), indicating the requirement of CLK2 kinase activity in the regulation of NF-κB activation. Additionally, the pulldown assay demonstrated that CLK2 but not CLK2^K192R was able to interact with p65 in vitro (Fig. 4F). Given that CLK2 is a dual-specificity protein kinase, we hypothesized that CLK2 interacted with p65 by phosphorylating p65. To validate this hypothesis, we performed an ADP-Glo™ kinase assay[44] and Phos-tag SDS–PAGE experiments to examine whether CLK2 induced p65 phosphorylation. The results showed that CLK2 increased the consumption of ATP in the presence of p65 in the kinase assay, while CLK2 but not CLK2^K192R resulted in a shift in p65 in Phos-tag SDS–PAGE, suggesting that CLK2 phosphorylated p65 in a manner dependent on its kinase activity (Fig. 4G and Supplementary Fig. 4c).

Next, we examined the biological role of p65 phosphorylated by CLK2. Posttranslational modifications can regulate protein stability and activity[45–47], and we examined the protein stability of p65 in CLK2-deficient cells; the results showed that CLK2 deficiency did not affect endogenous p65 stability (Supplementary Fig. 4d). Since p65 is shuttled from the cytosol to the nucleus to act as a transcription factor, we hypothesized that CLK2 phosphorylated p65 in the nucleus to reverse p65 nuclear translocation, thereby limiting the rapid accumulation of nuclear p65 and maintaining inflammatory and innate immune homeostasis. To determine whether the subcellular localization of p65 changed after virus stimulation in the absence of CLK2, we used HEK293T cells and found that p65 slowly transferred from the cytosol to the nucleus after SeV induction. However, CLK2 deficiency disrupted this balance, allowing p65 to quickly transfer into the nucleus after exposure to SeV for 12 h (Fig. 4H). Western blot analysis of the nuclear and cytosolic fractions revealed increased nuclear p65 levels and decreased cytosolic levels in CLK2-deficient cells after virus stimulation compared to those in wild-type cells (Fig. 4I). Furthermore, overexpression of CLK2 did not influence the cytosolic localization of p65 in resting cells but resulted in an exclusive reaction between

nuclear p65 and CLK2 after viral infection (Fig. 4J). These results suggest that CLK2 regulates the dynamic balance of p65 between the nucleus and cytosol under physiological conditions. Collectively, these data demonstrate that CLK2 regulates the dynamic shuttling of p65 between the nucleus and cytosol via phosphorylation.

## Phosphorylation of p65 at Ser180 by CLK2 completely blocks the transcriptional activity of nuclear NF-κB

To investigate the residues of p65 that are phosphorylated by CLK2, the PhosphoSitePlus online tool was used to predict 26 potential serine and threonine residues of p65 that could be phosphorylated by CLK2. Subsequently, Phos-tag was used to analyze the phosphorylation of p65 in the presence of CLK2. The results revealed that substituting alanine for serine 180, 205, 276, 281, or 316 or threonine 254 of p65 blocked CLK2-induced p65 phosphorylation (Supplementary Fig. 5a, b). Given that some of these sites may be upstream of the actual target sites, these residues were substituted with aspartic acid, which is a constitutive phosphorylation mimic, to determine the phosphorylation sites[32,48,49]. The Phos-tag results showed that the substitution of serine residues 180 and 316 with aspartate prevented CLK2-induced p65 phosphorylation, indicating that CLK2 phosphorylated p65 at serine 180 and 316 (Fig. 5A and Supplementary Fig. 5c).

Further analysis revealed that serine 180 was relatively conserved compared with serine 316 in fruit flies and humans (Fig. 5B). Next, we sought to investigate the function of serine 180 and 316 of p65 in NF-κB signaling. The NF-κB luciferase reporter assay showed that p65^S180D but not p65^S316D completely blocked wild-type p65-induced NF-κB luciferase reporter activation, suggesting that the phosphorylation of p65 serine 180 is necessary for NF-κB deactivation (Supplementary Fig. 5d). Interestingly, immunofluorescence experiments revealed that overexpression of the p65^S180A mutant led to the formation of punctate aggregates in the nucleus, and these aggregates colocalized with the PML body, which is involved in transcription (Supplementary Fig. 5e, f). These data indicated that the unphosphorylated p65^S180A mutant had higher transcriptional activity than wild-type p65, possibly due to its increased affinity for the PML body, which was contrary to p65^S180D.

We further assessed the transcriptional activity of p65^S180D in NF-κB signaling. We observed that SeV-induced NF-κB reporter activation and *IFN-β* mRNA expression were significantly decreased in the presence of p65^S180D in p65-deficient cells compared with wild-type p65 in p65-deficient cells (Supplementary Fig. 5d, g). Flow cytometry demonstrated that p65^S180D significantly altered the frequency of infected GFP-positive cells compared with that of wild-type p65 after VSV-GFP infection, indicating that p65^S180D weakened the antiviral response (Supplementary Fig. 5h). To further analyze the role of serine 180 of p65 under physiological conditions, we constructed p65^S180A/KI and p65^S180D/KI knock-in cell lines (Fig. 5C). NF-κB luciferase assays and real-time PCR showed that p65^S180D/KI but not p65^S180A/KI completely blocked SeV-induced NF-κB luciferase reporter activation and *IFN-β* and *IκBα* mRNA expression (Fig. 5D, E). In addition, viral replication was significantly increased in the p65^S180D/KI knock-in cell line compared to the wild-type cell line after VSV-GFP infection, as determined by flow cytometry (Fig. 5F). Moreover, we observed that p65^S180D/KI inhibited

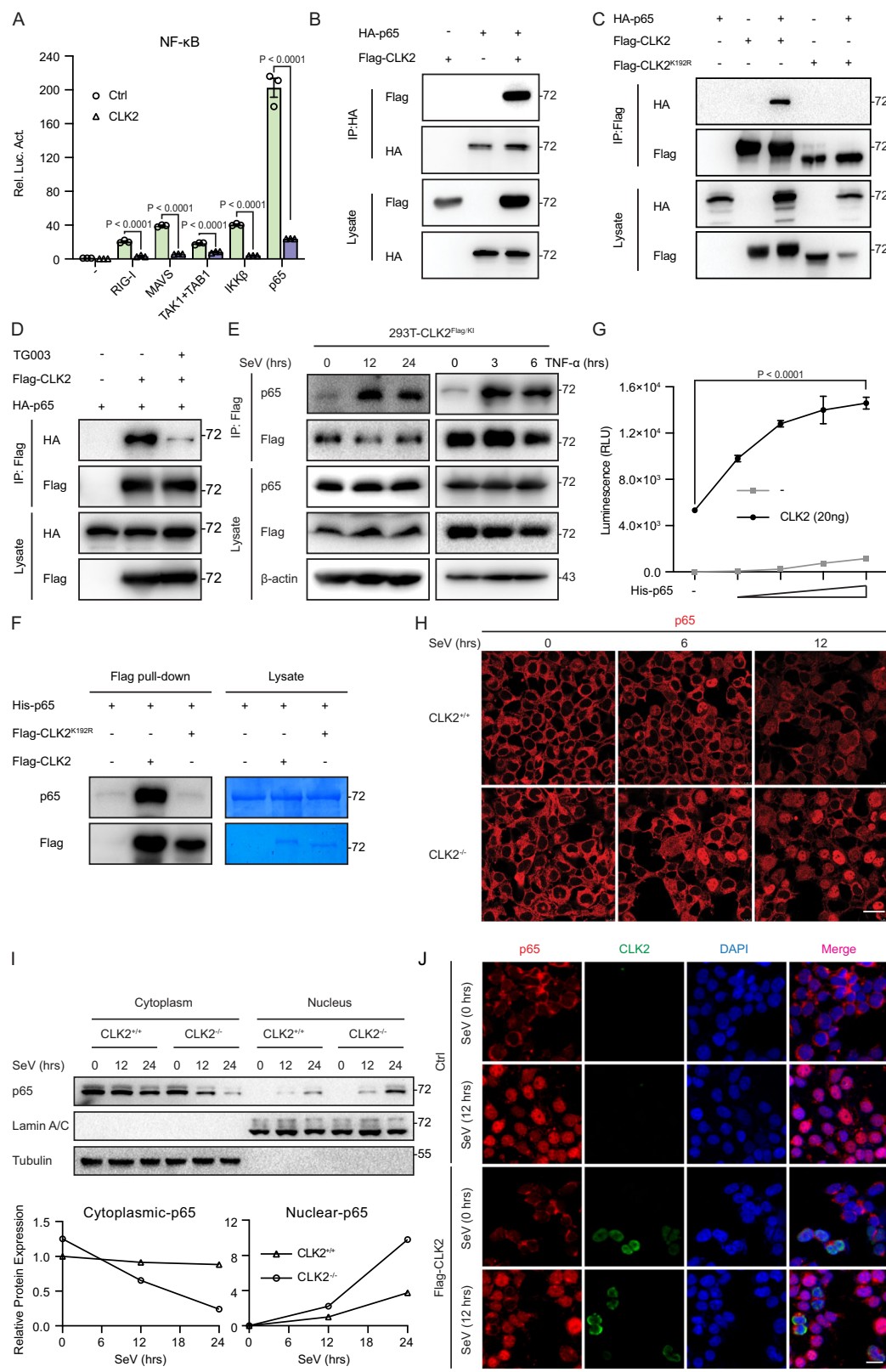

the expression of downstream genes induced by inflammatory cytokines. Similar to p65 deficiency, p65$^{S180D/KI}$ completely blocked TNF-α- or IL-1β-induced downstream gene production, but p65$^{S180A/KI}$ exhibited more powerful transcriptional activity (Fig. 5G and Supplementary Fig. 5i). Taken together, these results suggest that CLK2-mediated phosphorylation of p65 at serine 180 is essential for the deactivation of p65 and NF-κB signaling.

## Ser180 phosphorylation of p65 results in its degradation and nuclear export

Given that serine 180 phosphorylation of p65 terminates its activation, we next investigated whether the phosphorylation of p65 at serine 180 affected its transcriptional activity by affecting its expression or subcellular localization. p65$^{S180D/KI}$ cells exhibited lower p65 protein levels than wild-type p65 or p65$^{S180A/KI}$ cells, but there was no difference in p65

**Fig. 4 | CLK2 interacts with and phosphorylates p65 to reverse its nuclear localization. A** Luciferase reporter experiments analyzing NF-κB activity in HEK293T cells transfected with plasmids encoding RIG-I, MAVS, TAK1/TAB1, IKKβ and p65 plus CLK2 for 24 h ($n = 3$). **B, C** The interaction between CLK2 and p65 was determined by coimmunoprecipitation with the indicated antibodies and immunoblot analysis of HEK293T cells transfected with HA-p65, Flag-CLK2 or Flag-CLK2$^{K192R}$. **D** The interaction between CLK2 and p65 by coimmunoprecipitation with the indicated antibodies and immunoblot analysis of HEK293T cells transfected with HA-p65 and Flag-CLK2 with or without TG003 treatment (20 μM). **E** Endogenous coimmunoprecipitation was performed on CLK2$^{Flag/KI}$ HEK293T cells in the presence or absence of SeV and TNF-α stimulation for different times. **F** An in vitro pulldown assay was performed with Flag-CLK2 and His-p65, and the proteins were examined by western blotting. **G** The ADP-Glo$^{TM}$ kinase assay (Promega) confirmed the kinase activity of CLK2 and His-p65 in vitro, and the luminescence

was examined by a Glo-Max 20/20. **H** Immunofluorescence analysis of the localization of endogenous p65 (red) in CLK2$^{+/+}$ and CLK2$^{-/-}$ HEK293T cells infected with SeV at the indicated time points. Images were obtained by fluorescence microscopy. The scale bar is 10 μm. **I** Cytoplasmic and nuclear fractions of CLK2$^{+/+}$ and CLK2$^{-/-}$ HEK293T cells infected with SeV at the indicated time points before immunoblot analysis with anti-p65, anti-Tubulin and anti-LaminA/C antibodies. **J** Immunofluorescence analysis of the localization of endogenous p65 (red) in the presence or absence of Flag-CLK2 in HeLa cells infected with SeV at the indicated time points. Images were obtained by fluorescence microscopy. The scale bar is 20 μm (Ctrl: vector plasmid control). The data are representative of three independent experiments. The data are presented as the means ± SEMs. Statistical significance was analyzed by two-tailed Student's t test (**G**) or two-tailed ANOVA (**A**) (****$p < 0.0001$). Source data (**A**–**G**, **I**) are provided as a Source Data file.

mRNA levels. This result suggested that continuous phosphorylation of p65 at residue 180 leads to its protein degradation (Supplementary Fig. 6a, b). Notably, CLK2 overexpression decreased the protein levels of p65 in a dose-dependent manner (Supplementary Fig. 6c), confirming that the phosphorylation of p65 by CLK2 resulted in its degradation. Further analysis using proteasome and lysosomal inhibitors revealed that the degradation of p65$^{S180D/KI}$ could be rescued by MG132 but not by NH$_4$Cl (Supplementary Fig. 6d), and fractionation of the nucleus and cytosol revealed that p65$^{S180D}$ was degraded in the nucleus and that IκBα was depleted and never resynthesized (Supplementary Fig. 6e), suggesting that the degradation of p65$^{S180D}$ was dependent on the ubiquitin–proteasome system and occurred in the nucleus. Moreover, nuclear p65$^{S180D}$ exhibited higher ubiquitination levels than wild-type p65 (Supplementary Fig. 6f), and a cycloheximide (CHX) assay revealed that p65$^{S180D}$ had a significantly shorter half-life than p65$^{WT}$. On the contrary, p65$^{S180A}$, which is similar to p65$^{WT}$ in CLK2$^{-/-}$ cells, showed a bit increased stability (Supplementary Fig. 6g). Collectively, these results demonstrated that the phosphorylation of p65 at serine 180 led to its degradation in the nucleus, which was dependent on the ubiquitin–proteasome pathway.

We observed no difference in the expression of p65 between the CLK2-deficient cell line and the wild-type cell line (Supplementary Fig. 4d), and the degradation induced by the p65 S180D mutation may be due to artificial changes. Moreover, we examined the nuclear localization of p65, which was dramatically affected by CLK2 deletion in response to stimuli (Fig. 4H, I). To further investigate whether the localization of p65 was affected in p65$^{S180}$ mutant knock-in cells, immunofluorescence (IF) experiments were performed to evaluate the localization of p65$^{S180D/KI}$ in response to SeV or TNF-α stimulation. Following SeV stimulation, the majority of wild-type p65 was located in the nucleus, while almost all of the p65$^{S180D/KI}$ mutant was located in the cytoplasm, which was significantly distinct from the distribution of wild-type p65 (Fig. 6A). To confirm the hypothesis that active nuclear p65 phosphorylation led to its cytoplasmic redistribution, we examined the localization of p65 from the early stage to the later stage after TNF-α stimulation by immunofluorescence analysis. p65 entered the nucleus as early as 15 min and remained in the nucleus for approximately 3 h after TNF-α stimulation in wild-type cells. However, the nuclear localization of the p65$^{S180D/KI}$ mutant was delayed, and the p65$^{S180D/KI}$ mutant rapidly redistributed to the cytoplasm (Fig. 6B). Additionally, p65$^{S180A/KI}$ exhibited extended nuclear localization, which was consistent with our hypothesis (Fig. 6B). These results are consistent with the effect of CLK2 on dynamic changes in p65 sublocalization, suggesting that CLK2 regulates the subcellular redistribution of p65 by phosphorylating Ser180 of p65.

We next examined the mechanism underlying the ability of CLK2 to reverse p65 nuclear localization via phosphorylation of the Ser180 residue. We first evaluated the nuclear export of p65 and hypothesized that the phosphorylation of p65 at serine 180 resulted in nuclear export. To verify this hypothesis, we performed coimmunoprecipitation experiments to assess the binding affinity between p65, as well as

the p65$^{S180}$ mutant and the nuclear export protein CRM1 or IκBα. After normalizing the results to total p65, we found that p65$^{S180D/KI}$ did not interfere with the binding of p65 to p50 but dramatically increased the binding to nuclear exportin protein CRM1 (Fig. 6C). Considering that IκBα cannot be resynthesized in p65$^{S180D/KI}$ cells, an exogenous coimmunoprecipitation experiment was performed and showed that p65$^{S180D}$ had higher affinity for IκBα (Supplementary Fig. 6h). Moreover, p65$^{S180A}$ and p65$^{WT}$ in CLK2$^{-/-}$ cells exhibited decreased interactions with CRM1 and IκBα (Fig. 6D and Supplementary Fig. 6i), which was consistent with the results of p65$^{S180D}$ and indicated that Ser180 phosphorylation promoted p65 binding with CRM1 and IκBα to shuttle from the nucleus to the cytoplasm. Interactions with IκBα are typically associated with DNA-binding activity, so we next examined whether p65$^{S180D/KI}$ affected p65 binding to DNA. ChIP–qPCR assays revealed that p65$^{S180D/KI}$ nearly lost the ability to bind DNA; in contrast, p65$^{S180A/KI}$ exhibited increased DNA-binding activity (Fig. 6E). In addition, p65 in CLK2$^{-/-}$ cells exhibited increased DNA binding (Supplementary Fig. 6j). In summary, the phosphorylation of p65 by CLK2 at Ser180 decreases the DNA-binding activity of p65 and enhances its interaction with CRM1 and IκBα, resulting in the export of p65 from the nucleus.

To determine whether endogenous phosphorylation of the p65 Ser180 residue occurred, we produced a Ser180 phosphorylation antibody and verified it in wild-type and p65-deficient cells by examining SeV- or TNF-α-induced p65 activation. The results showed that phosphorylation of the serine 180 residue occurred and that it dynamically changed in response to SeV or TNF-α stimulation, and the specificity of the antibody was determined in p65$^{-/-}$ cells (Supplementary Fig. 6k). Furthermore, we found that the level of Ser180 phosphorylation increased in the early stage and decreased in the later stage after stimulation in wild-type cells but not in CLK2-deficient cells, suggesting that there was a physiologically dynamic change in Ser180 phosphorylation after exposure to viruses and inflammatory cytokines (Fig. 6F). Ser536 phosphorylation and Lys310 acetylation are required for full transcriptional activity of p65, and Ser468 phosphorylation negatively regulates transcriptional activity but can also be considered a hallmark of p65 activation[32,34,50]. To determine the relationships between Ser180 and Lys310, Ser468 or Ser536 modifications, we determined the levels of Ser180, Lys310, Ser468 and Ser536 of p65 in CLK2-deficient and p65$^{S180D/KI}$ cells in response to SeV and TNF-α stimulation. Consistent with the findings in primary cells (Fig. 2E and Supplementary Fig. 2f), CLK2 deficiency increased Ser468 and Ser536 phosphorylation and Lys310 acetylation (Fig. 6F), while p65$^{S180D/KI}$ decreased Ser468 and Ser536 phosphorylation and Lys310 acetylation. These results suggested that phosphorylation at Ser180 controls the posttranslational modifications of these sites (Fig. 6G). In addition, TG003 inhibited Ser180 phosphorylation and significantly increased Ser536 phosphorylation and Lys310 acetylation in HEK293T cells (Supplementary Fig. 6l), while cannot promote Ser536 phosphorylation and Lys310 acetylation in CLK2$^{-/-}$ cells (Supplementary Fig. 6m).

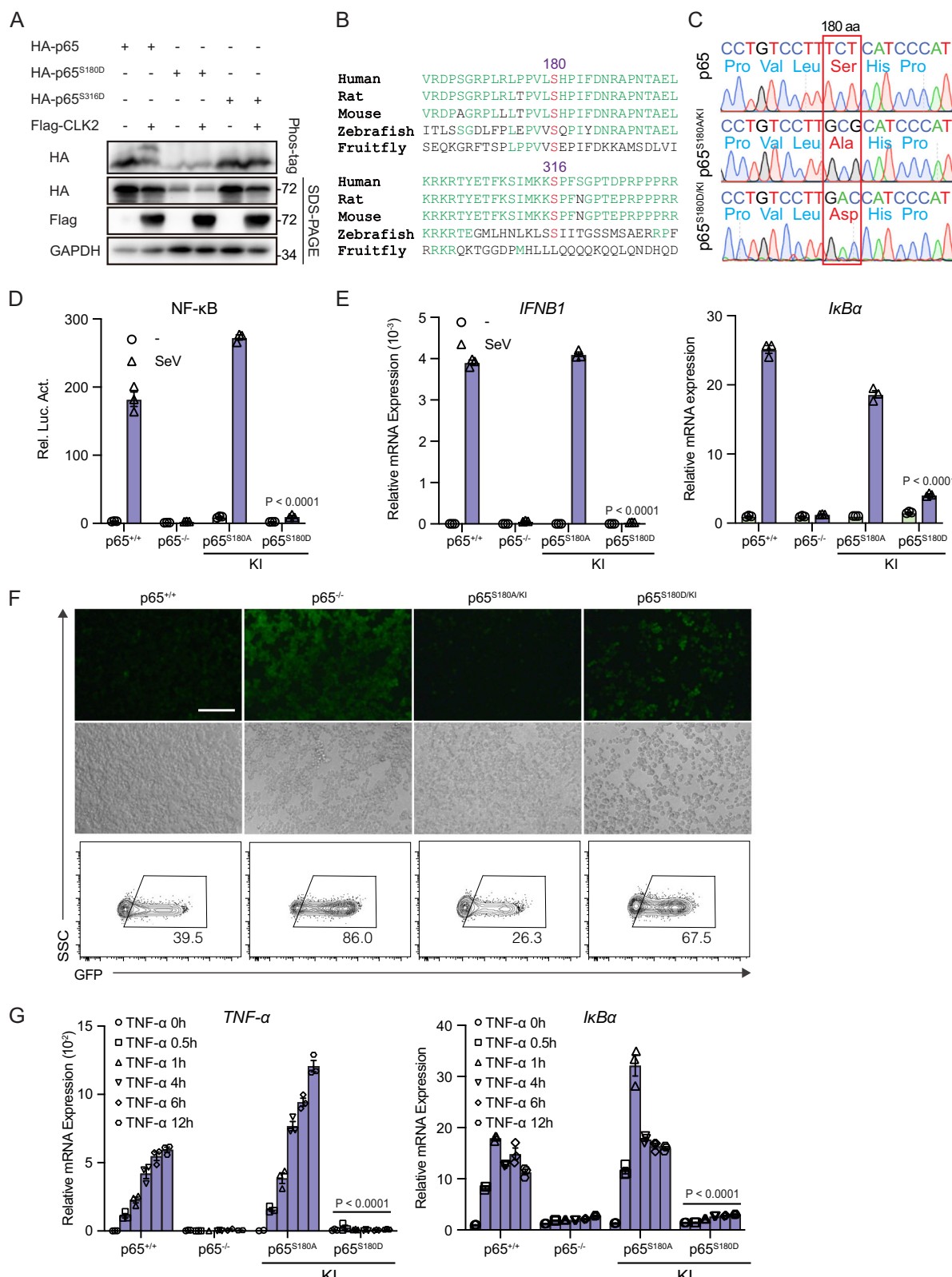

Taken together, the results obtained from primary cells and cell lines indicated that the phosphorylation of p65 by CLK2 acted as a basal checkpoint to prevent continuous activation of p65 and thus inhibit the activation of NF-κB.

**The CLK2 inhibitor TG003 exerts effects in vitro and in vivo**

To assess the therapeutic potential of targeting CLK2 during viral infection, TG003 was used to confirm its role in inflammation and the antiviral response. Its effects on the antiviral response were analyzed at the cellular level. Flow cytometry and plaque assays showed that TG003 effectively ameliorated VSV-GFP infection in THP-1 and HEK293T cells (Fig. 7A, B and Supplementary Fig. 7a, b). Real-time qPCR showed that the mRNA levels of IFNB1 and RANTES were markedly increased in TG003-treated cells compared to DMSO-treated cells after SeV infection (Fig. 7C and Supplementary Fig. 7c). To further verify the antiviral effects of TG003 in vivo, mice were

**Fig. 5 | The phosphorylation of p65 at Ser180 by CLK2 inhibits the transcriptional ability of activated NF-κB. A** Phos-tag SDS–PAGE analysis of the phosphorylation of the p65 residues 180 and 316 in the presence or absence of CLK2 in HEK293T cells. The upper shifted band represents the phosphorylated p65 protein. **B** Conserved site analysis of the p65 residues 180 and 316 from fruit flies to humans. **C** Identification of p65 knock-in cell lines by sequencing. **D** Luciferase reporter experiments analyzing NF-κB activity in the indicated cells infected with SeV for 24 h (*n* = 3). **E** Real-time PCR analysis of *IFNB1* and *IκBα* mRNA levels in the indicated cells infected with SeV for 12 h (*n* = 3). **F** The effects of p65 mutants on VSV-GFP infection for 12 h before phase contrast and fluorescence microscopy and flow cytometric analysis in the indicated cells. The scale bar is 250 μm. **G** Real-time PCR analysis of *IκBα* and *TNF-α* mRNA levels in cells exposed to 20 ng/ml TNF-α for 0 h. 0.5, 1, 4, 6, and 12 h (*n* = 3). The data are representative of three independent experiments. The data are presented as the means ± SEMs. Statistical significance was analyzed by two-tailed ANOVA (**D**, **E**, **G**) (****$p < 0.0001$). Source data (**A**–**G**) are provided as a Source Data file.

intraperitoneally injected with VSV in the presence or absence of TG003. Interestingly, TG003 could ameliorate the symptoms of VSV-induced conjunctivitis (Fig. 7D) and protect against VSV-induced death (Fig. 7E). Viral titers and antiviral cytokine levels in mouse sera were significantly reduced and increased, respectively, in TG003-treated mice after the injection of wild-type VSV (Fig. 7F, G). These results suggest that TG003 can improve antiviral activity in vivo and in vitro, making it a potential drug candidate for the treatment of viral infections.

Imiquimod (IMQ)-induced inflammatory lesions, which resemble human psoriasis, depend on the IL23/IL17 axis[51,52]. To explore the potential of CLK2 as a target in inflammatory diseases, an IMQ-induced psoriasis model was generated in CLK2-deficient mice or mice that were treated with TG003. The results showed that IMQ-induced ear swelling was increased in CLK2-deficient mice and mice that were intraperitoneally injected with TG003 (Fig. 7H). Moreover, we determined the tissue levels of inflammatory cytokines, including IL17A and IL17F. The mRNA and protein levels of psoriasis markers were significantly increased in CLK2-deficient mice and mice that were intraperitoneally injected with TG003 (Fig. 7I and Supplementary Fig. 7d). These data suggest that CLK2 deficiency or treatment with the CLK2 inhibitor TG003 can promote IMQ-induced inflammatory lesions in vivo, indicating that CLK2 may be a therapeutic target for inflammatory diseases and that inhibiting CLK2 may be a strategy to cure viral infection.

## Discussion

The transcription factor p65 is critical for cytokine-induced production of inflammatory genes and pathogen-induced expression of immune genes and is involved in the regulation of a variety of biological processes[53,54]. To but, the activation of p65 has been extensively studied, and the termination mechanism has been largely overlooked, since the majority of focus has been placed on IκBα-dependent protein synthesis in the later stage of activation[17]. However, little is known about how active nuclear p65 is rapidly repressed in the early stage of activation. It was reported that deacetylation and ubiquitination are involved in the rapid termination of p65 activation[30,31,55–57]. Here, we present an IκBα-independent rapid termination of active nuclear p65, which, when combined with the classic IκBα-dependent mechanism, forms a comprehensive negative regulatory system of NF-κB.

Whether the phosphorylation of p65, which acts as a checkpoint, regulates signaling and is involved in the termination of active p65 and nuclear export is largely unknown. We aimed to identify kinases that participate in antiviral and inflammatory homeostasis to identify the mechanism that terminates p65 transcriptional activation. Our previous research revealed that NLK phosphorylated MAVS for degradation to negatively regulate the antiviral innate immune response[37]. Furthermore, we found that CLK2 could block SeV-induced type I interferon signaling activation. We further confirmed that CLK2 inhibited NF-κB signaling, thereby suppressing inflammation and innate immunity. Moreover, mouse experiments revealed that Clk2 deficiency promoted inflammatory and antiviral responses, which was consistent with the in vitro results. Furthermore, we investigated the molecular mechanisms by which CLK2 inhibited inflammatory and antiviral responses by mediating the degradation and nuclear export

of active nuclear p65 through the phosphorylation of p65 at the Ser180 residue. These findings highlight the role of CLK2 in regulating inflammatory and antiviral responses, suggesting that CLK2 may be a promising target for treating inflammation-associated diseases and viral infections. To test this hypothesis, we administered the CLK2 inhibitor TG003 to human cells and mice and observed clear effects on viral infection in vitro and in vivo and the promotion of IMQ-induced inflammatory lesions in vivo.

A strong inflammatory and innate immune response can protect the host from pathogens and death within a certain period of time. However, persistent or excessive inflammatory and innate immune responses, such as cytokine storms, can lead to the host expending extra energy, which is harmful to the health and lifespan of individuals[58–63]. Evolutionarily, hosts have developed a precise balance between long-term and short-term survival, which is achieved through strong negative regulation[64]. The negative feedback loop of p65 is regulated at the transcriptional level via the production of excess IκBα. The posttranslational step involves acetylation and may be involved in the activation and regulation of p65[65,66]. Additionally, phosphorylation has been reported to regulate the activation of p65[67,68]; however, the roles of phosphorylation in negative regulation in the early stage remains poorly defined. Here, we identified p65 as a substrate of CLK2. The phosphorylation of p65 at Ser180 by CLK2 resulted in its degradation and nuclear export. Compared with wild-type p65, the p65$^{S180D}$ mutant, which was a persistent phosphorylation mimic, was resistant to nuclear localization and had reduced transcriptional activity. These results suggest that CLK2 plays an early role in inflammatory and innate immune responses by phosphorylating nuclear p65 at Ser180. The phosphorylation of p65 at Ser180 terminates active p65 through degradation and nuclear export. These results indicate that CLK2 acts as a brake in the early stage of NF-κB transcriptional activation and works in conjunction with IκBα, which functions as a brake in the later stage, to limit transcription throughout NF-κB signaling activation. Furthermore, CLK2-mediated phosphorylation of p65 at Ser180 results in homeostatic suppression of p65 activity before activation of the signaling cascade.

In particular, we discovered a phosphorylation site on p65: Ser180. Through the use of site-specific phosphorylation antibodies, we demonstrated that Ser180 was phosphorylated under physiological conditions and was dynamically altered in response to viral or cytokine stimulation. We further showed that Ser180 phosphorylation of p65 by CLK2 could inhibit signaling pathways induced by a variety of PAMPs and cytokines, including RNA viruses, DNA viruses, LPS, IL-1β and TNF-α, suggesting that Ser180 phosphorylation may be a universal inhibitory mechanism for inflammatory and innate immune responses to a wide range of stimuli.

The expression levels of p65$^{S180D}$ were significantly lower than those of wild-type p65, and p65$^{S180D}$ was degraded through the ubiquitin–proteasome system in the nucleus. The CHX assay showed that p65$^{S180D}$ had a shorter half-life than wild-type p65. Previously, it was reported that nuclear p65 was degraded by the PDLIM2 E3 ligase[30]. Our results revealed that the phosphorylation of p65 at Ser180 was an upstream switch for p65 ubiquitination. Under physiological conditions, no significant degradation of p65 was observed, possibly due to regulation at the transcriptional and translational levels. Moreover, in contrast to the inactivation of p65$^{S180D}$, p65$^{S180A}$ exhibited increased

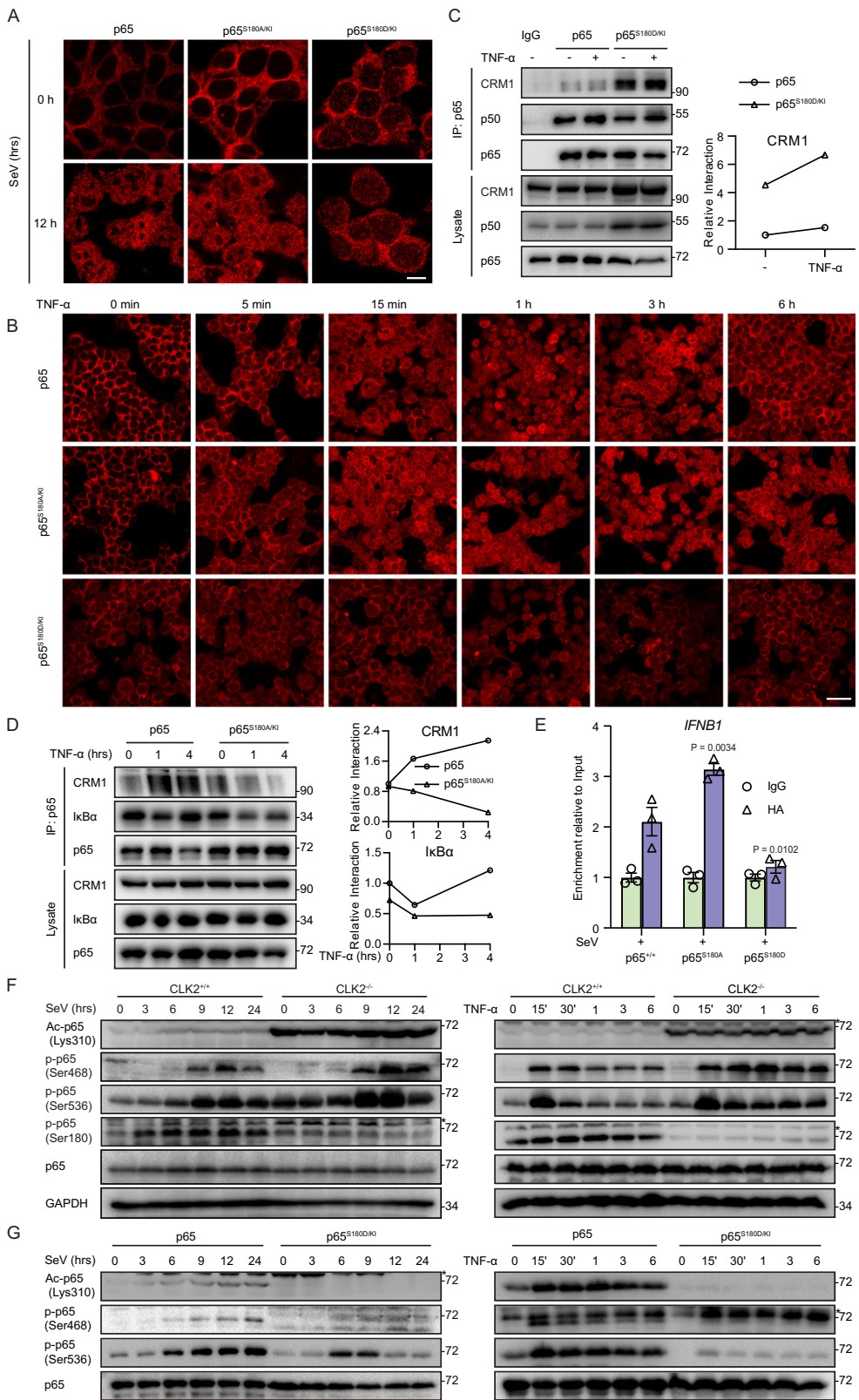

transcriptional activity and increased resistance to viral infection. p65$^{S180A}$ interacts with the PML body and may participate in the formation of a hypertranscriptional activation complex to regulate transcriptional activity[69]. Additionally, p65$^{S180A}$ and p65 in CLK2$^{-/-}$ cells showed reduced affinity for CRM1 and IκBα and increased DNA-binding activity, further demonstrating how Ser180 phosphorylation affects transcription. These results demonstrate the IκBα-independent

negative regulatory mechanisms of p65, thereby advancing our understanding of the NF-κB signaling pathway.

In addition to the Ser180 residue, the Phos-tag experiments demonstrated that Ser316 was also phosphorylated by CLK2. Our findings suggest that Ser316 may not play a significant role in the transcriptional activation of NF-κB, which is consistent with the findings of Akihide Ryo[70]. However, Wang et al. reported that IL-1β induced

**Fig. 6 | Ser180 phosphorylation of p65 results in its degradation and nuclear export. A** Immunofluorescence analysis of the localization of endogenous p65 (red) in p65[+/+], p65[S180A/KI] or p65[S180D/KI] HEK293T cells infected with SeV at the indicated time points. Images were obtained by fluorescence microscopy. The scale bar is 10 μm. **B** Immunofluorescence analysis of the localization of endogenous p65 (red) in p65[+/+], p65[S180A/KI] or p65[S180D/KI] HEK293T cells exposed to TNF-α at the indicated time points. Images were obtained by fluorescence microscopy. The scale bar is 40 μm. **C** Coimmunoprecipitation of endogenous p65 and p50, IκBα or CRM1 in in p65[+/+] or p65[S180D/KI] HEK293T cells in the presence or absence of TNF-α stimulation for 1 h. **D** Coimmunoprecipitation of endogenous p65 and IκBα or CRM1 in p65[+/+] or p65[S180A/KI] HEK293T cells in the presence or absence of TNF-α stimulation. **E** ChIP assays were performed on p65[+/+], p65[S180A/KI] and p65[S180D/KI] HEK293T cells stimulated

with SeV, and the promoter of IFNB1 was examined by real-time qPCR (n = 3). The sequences are presented in Supplementary Table 2. **F** Western blot analysis of phosphorylated Ser180, Ser468, and Ser536 p65; acetylated Lys310 p65; total p65; and GAPDH in CLK2[+/+] and CLK2[−/−] HEK293T cells exposed to SeV or TNF-α for the indicated times. The upper band marked with asterisk indicates the unspecific band. **G** Western blot analysis of phosphorylated Ser468 and Ser536 p65, acetylated Lys310 p65 and total p65 in wild-type and p65[S180D/KI] 293T cells infected with SeV and TNF-α at the indicated times. The data are representative of three independent experiments. The upper band marked with asterisk indicates the unspecific band. The data are presented as the means ± SEMs. Statistical significance was analyzed by two-tailed ANOVA (**E**) (*p < 0.05, **p < 0.01). Source data (**C**–**G**) are provided as a Source Data file.

the phosphorylation of p65 at Ser316, leading to p65 activation[71]. The specific role of Ser316 in NF-κB signaling activation remains unclear. Although we did not observe an effect of phosphorylation at the serine 316 residue on p65 activity, our results confirmed that CLK2 was a kinase that phosphorylated the serine 316 residue of p65. Thus, further experiments are necessary to determine the biological function of the p65 serine 316 residue. Following the conserved site analysis of the p65 residues 180 and 316 across species from fruit flies to humans, we attempted to identify the consensus amino acid sequence of p65 that was targeted by CLK2. The PROMALS3D online tool was used and yielded no significant results. Given that Ser50 of PTP1B is the sole site that has been shown to be targeted by CLK2 and that no additional substrates can be concurrently analyzed, the determination of this target remains challenging. However, further investigation may eventually identify this target. Furthermore, our findings revealed that the phosphorylation of residues 205, 276, 281 and 254 is required for the phosphorylation of residues 180 and 316. Notably, these 4 residues are located in the dimerization domain of p65, which may be associated with p65/p50 nuclear localization. Therefore, we postulate that phosphorylation of these 4 sites precedes the nuclear localization of p65/p50 and the subsequent phosphorylation of serines 180 and 316 in the nucleus. Further analysis of the relationship between the phosphorylation of these 4 residues and dimerization is warranted.

The protein levels of CLK2 exhibit dynamic changes at different time points after stimulation, while the mRNA levels show slight alterations, suggesting the involvement of posttranslational modifications in the regulation of CLK2 activity. However, further exploration is required to understand the regulatory mechanism of CLK2 activity in response to viruses or cytokines. In addition, our findings suggested that the loss or inactivation of CLK2 could lead to the subsequent activation of p65 and chronic inflammation, thereby contributing to the development of cancer and autoimmune diseases. Therefore, the function of p65[S180D] in vivo should be further validated using p65[S180D] knock-in mice. On the other hand, CLK2 can be regulated significantly by SeV, TNF-α or other stimulations. Activated CLK2 is usually considered to be facilitate RNA alternative splicing processes involving SRSF1 and RBFOX2. When CLK2 is activated by infection to target p65, it may also promote the activation of SRSF1, which has been found that overexpression can inhibit the production of pro-inflammatory cytokines[72]. However, the promotion of RBFOX2, which can be regulated by CLK2, is considered to be positively associated with inflammation[73]. In summary, these findings underscore the complex regulatory role of CLK2 in the inflammatory response, potentially exhibiting dual effects in different inflammatory diseases or virus infection. Moreover, a cytokine storm, which is an uncontrolled and excessive response driven by NF-κB and various signaling pathways, is lethal for living organisms[62,63]. Consequently, it is imperative to determine the impact of CLK2 on cytokine storms, since CLK2 may represent a therapeutic target in the future.

Based on these results, we proposed a model for the role of CLK2 in inflammatory and innate immune responses. PAMPs and cytokines induced changes in the subcellular localization of p65, which disrupted

the dynamic balance between the cytosol and nucleus. Once p65 translocates into the nucleus, CLK2 interacts with and phosphorylates p65 at Ser180, thereby removing activated PTMs, reversing its nuclear localization and mediating the degradation of a portion of nuclear p65. This effectively restricts the speed of nuclear import of p65 and terminates constitutive transcriptional activation in the early stage (Fig. 8). Moreover, CLK2 reinforces the inhibition of NF-κB by increasing its protein expression in response to stimulation.

In summary, our study revealed the negative regulatory role of CLK2 in NF-κB-mediated inflammatory and innate immune responses. We also identified a kinase and phosphorylation site of p65 and determined the mechanism underlying the rapid termination of nuclear p65 activation. These findings describe the early termination of nuclear NF-κB, complementing the IκBα-dependent negative feedback model and completing the negative regulation system associated with NF-κB activation. Thus, these findings may provide a therapeutic target for treating inflammation-related diseases and cytokine storms.

## Methods

### Ethics statement

All animal studies were conducted in accordance with the Guidelines of the China Animal Welfare Legislation and approved by the Committee on Ethics in the Care and Use of Laboratory Animals of Wuhan University (permit number: 15060A). All efforts were made to minimize suffering.

### Reagents and constructs

The antibodies and reagents used in the study are presented in Supplementary Table 1. Plasmids encoding CLK2, p65 and the mutants were constructed according to standard molecular cloning techniques. The sequences of the DNA ligands and primers used are listed in Supplementary Table 2. The IFN-β, NF-κB and ISRE (specific sequence: 5′-AGGGAAAGTGAAACT-3′) luciferase reporter plasmids and SeV, VSV and VSV expressing GFP (VSV-GFP) were gifts from Hong-Bing Shu and Bo Zhong[74–76] (Wuhan University, Hubei, China).

### Cell lines and mice

HEK293T, HeLa, THP-1 and Vero cell lines were originally obtained from the ATCC. CLK2[−/−] HEK293T cells were generated using CRISPR/Cas9-mediated gene knockout. The sgRNA sequences are presented in Supplementary Table 2. The p65[−/−] HEK293T cell line was a gift from Bo Zhong (Wuhan University, Hubei, China). For p65[S180A/D] knock-in cell lines, a homology-directed repair (HDR)-based strategy was used. First, the gRNA sequence and PAM motif with high scores near the Ser180 residue of p65 were chosen. Then, the gRNA sequence was cloned and inserted into the lentiCRISPRv2 vector, which was subsequently transfected with a single-stranded DNA template containing the Ser180A/D mutation into 293T cells. The colonies were screened by genomic PCR with screening primers and then sequenced. For CLK2 flag-tagged knock-in cell lines, the most specific PAM motif closest to the CLK2 termination codon was chosen and subsequently inserted into the lentiCRISPRv2 vector. Moreover, the homologous arms,

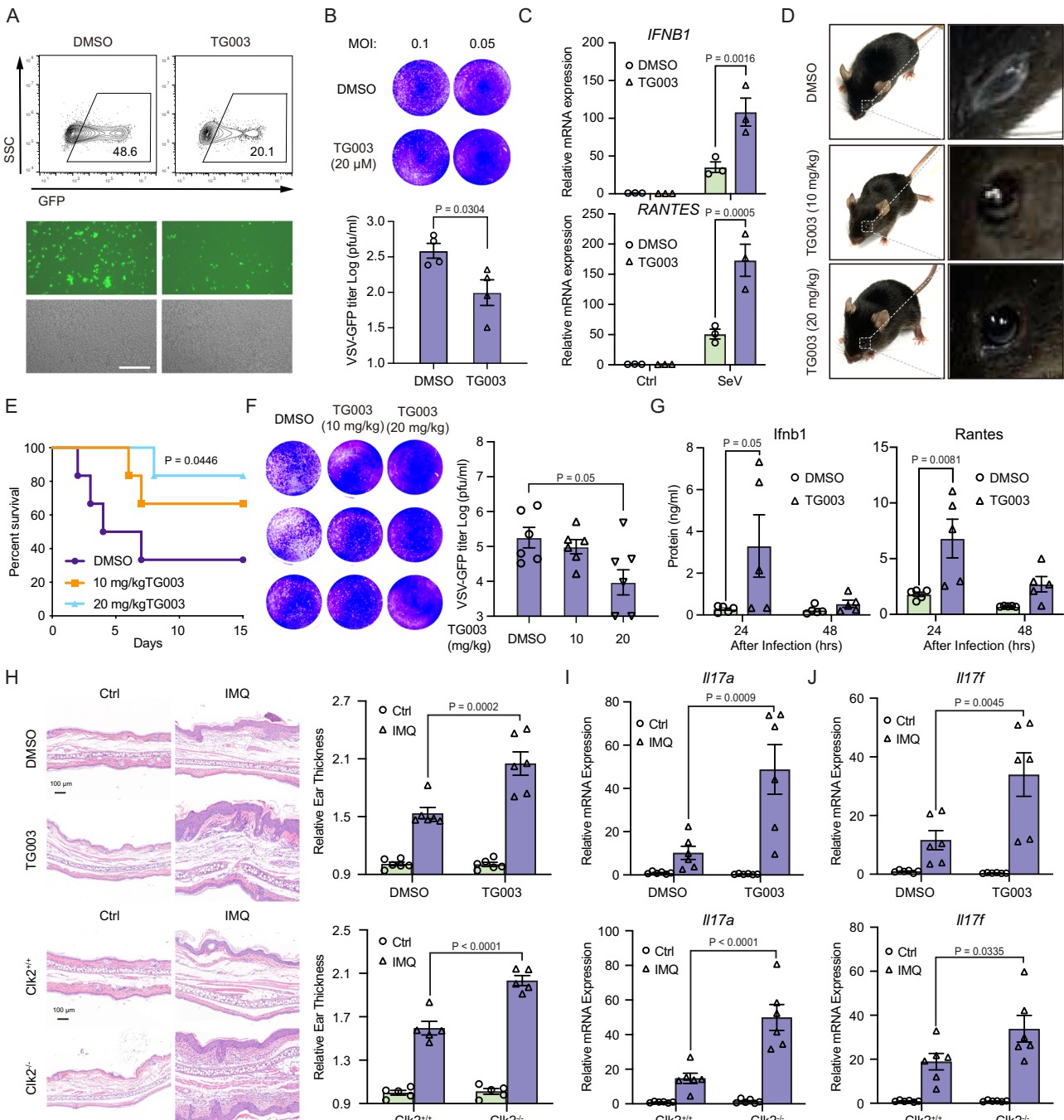

**Fig. 7 | The CLK2 inhibitor TG003 exerts effects in vitro and in vivo. A** THP-1 cells were infected with VSV-GFP at an MOI of 1 for 12 h in the presence or absence of TG003 (20 μM) before phase contrast and fluorescence microscopy and flow cytometric analysis. The scale bar is 250 μm. **B** THP-1 cells were infected with VSV-GFP at an MOI of 1 for 1 h in the presence or absence of TG003 (20 μM) before the medium was replaced, and the cells were cultured for 24 h. Then, the supernatants were diluted and added to Vero cells for plaque assays (*n* = 3). **C** Real-time PCR analysis of *IFNB1* and *RANTES* mRNA levels in the presence or absence of TG003 in THP-1 cells infected with SeV for 12 h (*n* = 3). **D** TG003 alleviated VSV-induced conjunctivitis at 10 mg/kg or 20 mg/kg. **E**–**G** The effects of TG003 on the antiviral response in vivo. 6-week-old female mice (*n* = 6) were infected with VSV ($2 \times 10^7$ pfu/g), after which

DMSO or 10 mg/kg or 20 mg/kg TG003 was intraperitoneally injected three times each day. **E** Mouse survival (Kaplan–Meier curve) was monitored for 15 days. **F** Serum samples were collected after 24 h and subjected to plaque assays (*n* = 6) and (**G**) ELISA analysis of Ifnb1 and Rantes (*n* = 5). **H** H&E staining and ear thickness in IMQ-treated mice on Day 6 (*n* = 6). **I, J** Real-time PCR analysis of *Il-17a* and *Il-17f* mRNA levels after treatment of Clk2-deficient mice with IMQ for 6 days or after intraperitoneal injection of TG003 (*n* = 6). The data are representative of three independent experiments. The data are presented as the means ± SEMs. Statistical significance was analyzed by two-tailed Student's *t* test (**B**, **F**) or two-tailed ANOVA (**C**, **G**–**J**) (**p* < 0.05, ***p* < 0.01, ****p* < 0.001, *****p* < 0.0001). Source data (**A**–**J**) are provided as a Source Data file.

including the left and right arms on the flank of the CLK2 termination codon, were cloned and inserted into the pAAV-FLAG-BIS vector. Then, the homologous arm and sgRNA were transfected into 293T cells, resulting in targeted double-strand breaks, which were repaired by HDR in the presence of a single-stranded DNA template containing the

Flag-Tag mutation. The Flag-tagged knock-in cell line was finally obtained by blasticidin screening and monoclonal screening. To acquire BMDMs and BMDCs, bone marrow cells were isolated from 6-week-old C57BL/6J Clk2 knockout mice (two male mice and two female mice), and 1×ACK buffer was used to lyse blood cells. Then, the bone

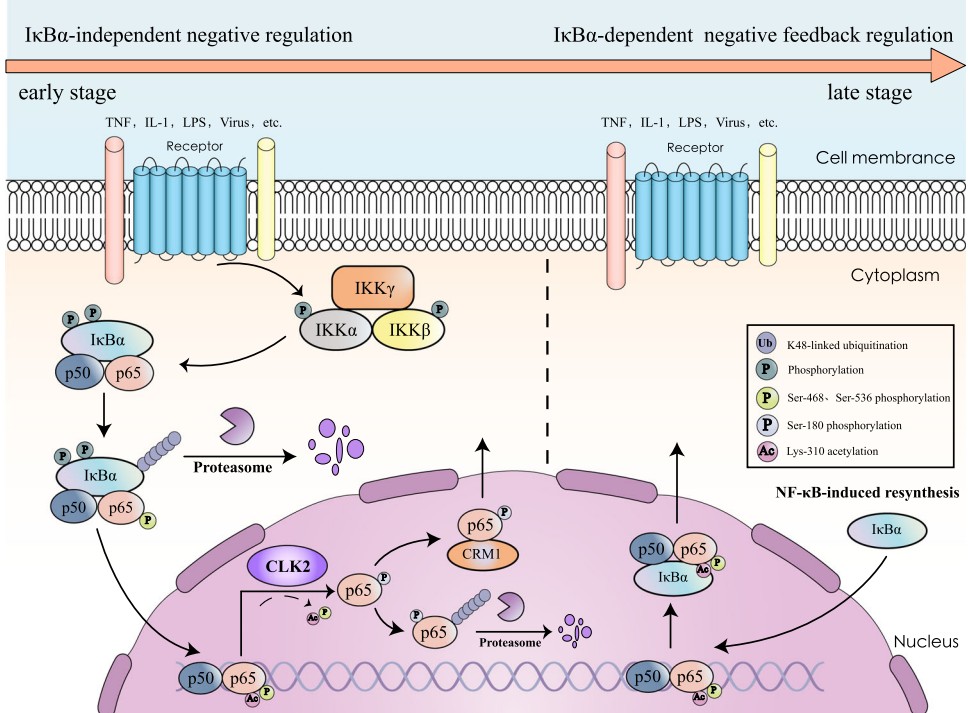

**Fig. 8 | Schematic model of CLK2-mediated IκBα-independent early termination of NF-κB activation.** Upon stimulation, NF-κB signaling induces the expression of multiple genes. Following activation, IκBα is phosphorylated and degraded, allowing the released p50/p65 dimer to translocate to the nucleus to mediate the transcription of downstream genes. In the early stage of NF-κB activation, this process is rapidly inhibited by CLK2 through the phosphorylation of p65, leading to its degradation and cytoplasmic redistribution. In the late stage, resynthesized IκBα reassociates with the nuclear NF-κB dimer, resulting in its export from the nucleus. Thus, these events enhance our understanding of the spaciotemporal termination of NF-κB activation.

marrow cells were incubated in DMEM containing 10% FBS and M-CSF (10 ng/ml) for BMDMs or GM-CSF (20 ng/ml) for BMDCs. After 3 days, the supernatant was renewed for further experiments. To acquire MLFs, Lungs were isolated from 10-week-old mice (two male mice and two female mice), and lungs were cut up in 1×HBSS buffer with 10 mg/ml type II collagenase and 20 μg/ml DNase I. Then lung cells were digested for 4 h at 37 °C. Cell suspensions were cultured in DMEM containing 10% FBS[37]. C57BL/6J Clk2 knockout mice were purchased from GemPharmatech (B6/JGpt-*Clk2em37Cd5370*/Gpt stock no: T014676). The sequences of the primers used for genomic identification and the sgRNA sequences used for gene knockout are shown in Supplementary Table 2. Mice aged 6–10 weeks were used for the experiments and housed under a 12:12-h light/dark cycle at a controlled temperature. For the survival experiments, VSV or HSV-1 were injected into 6-week-old female mice via tail vein and intraperitoneally. To induce imiquimod-induced (IMQ) psoriasis-like skin inflammation, 8-week-old male mice were treated with 5 mg of 5% IMQ (Aldara; 3 M Pharmaceuticals) on the shaved back and right ear for 6 consecutive days. TG003 was intraperitoneally injected at a dose of 10 mg/kg three times per day for 3 consecutive days. On the 7th day, blood was collected from the heart, and the back skin and ears were removed and then incubated in 4% PFA. For sex and gender reporting, we employed equal number of male and female mice for primary cells isolation to reflect the general phenomenon in mice cells. Because female mice have less body weight, and less virus required, we used female mice in virus-induced survival experiments. The remaining male mice were utilized for imiquimod-induced (IMQ) psoriasis-like skin inflammation.

## Flow cytometry
The cells were infected with VSV-GFP at the indicated MOI for 12–16 h before being harvested, resuspended in phosphate-buffered saline (PBS), subjected to flow cytometry (Beckman Coulter, Fullerton, CA, USA) and analyzed using FlowJo software (TreeStar, Ashland, OR, USA).

## Plaque assays
The cells were infected with VSV or VSV-GFP at the indicated MOI for 1 h, washed three times with PBS and cultured in fresh medium for 12 h. The supernatants were harvested, diluted and used to infect confluent Vero cells for 1 h before being replaced with 1% methylcellulose. After 36 h, the methylcellulose was removed, and the cells were fixed with 4% formaldehyde for 30 min and stained with 0.25% crystal violet in 20% methanol. The plaques were counted, averaged and multiplied by the dilution factor to determine the viral titer as PFU/mL.

## Transfection and reporter assays
The cells were transfected using TurboFect reagent (Thermo Scientific, Waltham, MA, USA). For the reporter assay, the indicated luciferase reporters, pRL-TK *Renilla* luciferase, and different expression or control vectors were transfected in 24-well plates for 36 h before a dual specific luciferase reporter kit was used (Promega, Madison, WI, USA).

## Subcellular fractionation
The cells were lysed on ice for 20 min with hypotonic buffer (20 mM Tris-HCl 8.0, 1.5 mM $MgCl_2$, 10 mM KCl, 0.1% Triton X-100 and 20% glycerol). After centrifugation for 10 min at $1273 \times g$, the supernatants and pellets were used as the cytoplasmic fractions and nuclear fractions, respectively.

## Coimmunoprecipitation and immunoblot analyses
Transfected or stimulated cells were harvested in NP40 lysis buffer (50 mM Tris-HCL pH 8.0, 150 mM NaCl, and 0.5–1.0% NP40) supplemented with a protease inhibitor cocktail (Roche, Basel, Switzerland). After centrifugation at $12,580 \times g$ for 15 min, the supernatants were incubated with the indicated antibodies and Protein A or Protein G beads (Roche) at 4 °C overnight. The beads were then washed with NP40 wash buffer (50 mM Tris-HCL pH 8.0, 150 mM NaCl, and 0.1% NP40), and the immunoprecipitates were separated by SDS–PAGE. The

proteins were transferred to PVDF membranes (Bio-Rad, Hercules, CA, USA) for immunoblotting with the indicated antibodies. The uncropped and unprocessed scans of blots were exhibited in Source Data.

## Protein purification, pulldown and kinase assays

The His-p65 plasmid was constructed, and protein expression was induced with IPTG overnight at 18 °C. Lysozyme (1 mg/ml) was added, and the mixture was incubated on ice for 30 min. After centrifugation at $935 \times g$ for 15 min, the supernatant was incubated with His beads and then washed approximately three times. The proteins were eluted with elution buffer, and the protein concentration was determined by BCA. Flag-CLK2 and Flag-CLK2$^{K192R}$ were purified from transfected HEK293T cells, enriched by IP and then eluted with the Flag peptide. The His-p65 and Flag-CLK2 proteins were incubated in lysis buffer for 4 h at 4 °C and then washed with lysis buffer three times. Finally, the proteins were examined by SDS–PAGE. The His-p65 and CLK2 were purchased from Promega, used according to the ADP-Glo$^{TM}$ Kinase Assay (Promega) and detected by a Promega GloMax 20/20.

## RNA isolation and real-time PCR

Total RNA was isolated using TRIzol reagent (TaKaRa, Otsu, Japan) and reverse transcribed according to the manufacturer's instructions (Fermentas, Vilnius, Lithuania). The mRNA was quantified by real-time PCR. All real-time PCR values were normalized to those of *GAPDH*. The primers used in the study are listed in Supplementary Table 2.

## Immunofluorescence and confocal microscopy

The cells were fixed for 15 min with 4% paraformaldehyde and permeabilized for 15 min with 0.1% Triton X-100. The cells were then blocked with 5% bovine serum albumin (BSA) for 30 min and incubated overnight with primary antibodies in 5% BSA and then for 1 h with fluorescent secondary antibodies. DAPI was used for nuclear staining. Images were obtained with a confocal microscope.

## Statistical analysis

The data are expressed as the mean ± SEM. Statistical significance was evaluated using the unpaired two-tailed Student's *t* test, and comparisons among more than two groups were performed using ANOVA. Differences were considered significant at a *p* value < 0.05.

## Reporting summary

Further information on research design is available in the Nature Portfolio Reporting Summary linked to this article.

# Data availability

All data generated in this study are provided in the Source Data file. The source data underlying Figs. 1A–H, 2B–E, 3A, B, D, F, G, 4A–G, I, 5A, D, E, G, 6C–G, 7B, C, F–J and Supplementary Figs. 1a, c, d, 2a–f, 3a–d, g–j, l, 4b–d, 5a–d, g, i, 6a–m, 7a–d are available in the Source Data file whenever possible. Source data are provided with this paper.

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

## Acknowledgements
This work was supported by grants from the National Natural Science Foundation of China (32370777 for R.-L.D., 32270760 for S.-Z.L.) and the First Batch of High-level Talent Scientific Research Projects of the Affiliated Hospital of Youjiang Medical University for Nationalities in 2020. Professor Jun-Li Luo provided numerous suggestions throughout the paper writing process.

## Author contributions
S.-Z.L., Q.-P.S., H.-M.Z., Y.-Y.L., M.-Q.F., X.-Y.L., L.-Z.Q., Y.-N.H., X.-Y.L., X.-H.D., X.-C.H. and Y.-Z.C. performed the research. S.-Z.L. and Q.-P.S. analyzed the data. S.-Z.L. and Q.-P.S. wrote the paper. S.-Z.L., Q.-P.S., R.-L.D., Y.-X.L. and X.-D.Z. designed the experiments and discussed the data.

## Competing interests
The authors declare no competing interests.
