## [Peer Review File · Nature Communications]

CLK2 mediates I κ B α -independent early termination of NF- κ B activation by inducing cytoplasmic redistribution and degradationEditorial Note: Parts of this Peer Review File have been redacted as indicated to remove third-party material where no permission to publish could be obtained.

REVIEWER COMMENTS

Reviewer #1 (Remarks to the Author):

The major finding of this manuscript is the identification of the NF- κ B p65 subunit as a target of CLK2. Phosphorylation of p65 at Ser-180 leads to nuclear export and degradation. This finding uncovers a new substrate for CLK2, a kinase previously found to modify splicing factors. Most importantly this manuscript discovers a new mechanism for negative feedback of p65-driven gene expression that is proposed to be independent of the well-known p65-dependent production of the I κ B α inhibitor. The authors use complimentary approaches to demonstrate that loss of CLK2, CLK2 kinase function, or p65 Ser-180 phosphorylation leads to a gain of interferon and inflammatory responses that have functional impacts on virus control and inflammatory pathologies, in cell culture and in mouse models. The parsing of main figure and supplementary figures is appropriate. That being said, there are a few holes in their data that should be addressed to bolster CLK2 as the direct p65 kinase that functions prior to and independently of I κ B α .

Major issues:

1. They indicate that CLK2 is an early terminator of NF- κ B signaling via p65 on Ser180 and independent of I κ B α - but there are some discrepancies with this interpretation.
 - a. Most assays are at 6, 12, or 24 hrs post-stim, well-after I κ B α is typically transcriptionally induced by p65. Is there clear data showing two waves of inhibition? Does I κ B α ever function in the absence of CLK2 or does CLK2 dominate?
 - b. Does the phosphorylation change of p65 in response to CLK2 impact the ability of p65 to be bound by I κ B α ? If p65 is no longer recognized by I κ B α in the absence of CLK2 modifications then that could lead to an accumulation of p65 in the nucleus and increased transcriptional activity. p65 S180A is a mutant that can't be phosphorylated downstream of CLK2 and has a prolonged residence in the nucleus (Fig 6), but how is the role of I κ B α excluded? Do any of the phosphorylation site mutants impact targeting by I κ B α ?
2. Is CLK2 the sole kinase and what activates it?
 - a. They identify multiple phosphorylation sites but there is never an in vitro analysis with purified CLK2 protein. As it stands now- CLK2 is certainly an upstream kinase- but given IP is performed in HEK293T - CLK2 might be acting via another kinase? Does the CLK2 inhibitor TG003 block phosphorylation of Ser 180 on p65 or any other site?
 - b. In Fig 6D, the SeV infected CLK2^{-/-} have lower levels of phospho Ser180 p65 overall, but it is not eliminated. What kinase drives this remaining phosphorylation? What does the asterisk/upper band indicate in the western blots?
 - c. This CLK2-mediated inhibition is only observed upon stimulation- so what is activating CLK2?
3. There is an overinterpretation that PML-associated p65 is more transcriptionally active. The PML colocalization and phase separation data is not well developed, being performed in the context of overexpression- it raises more questions than answers. Consider removing.
4. A major flaw in the publication is the poor quality of writing. Professional editing is essential. There were numerous grammatical errors and improper word choices (e.g. 'we wonder CLK2 may be a pregnant target to treat inflammation-associated diseases and virus infection') that make it nearly impossible for the reader to work through. This did detract from my understanding of some of their summaries and interpretations.

Minor issues:

1. Fig 2B. The PCR genotyping data for WT vs KO mice seems swapped.
2. Supp Fig 3 and Fig 5. have an odd arrangement of figures where the letters are not in alphabetical order.
3. The font kerning for the sequence alignment of Fig 5B is off. A monospace font is needed.
4. Fig 6 Figure legend states data are presented as SEM- but that type of data is not in the figure.
5. Information on the strain and source of SeV, VSV, and HSV virus stocks is lacking.

Reviewer #2 (Remarks to the Author):

The manuscript entitled "CLK2-mediated IKappaBalpha-independent early-termination of NF-kappaB activation by phosphorylating nuclear-p65 for cytoplasmic redistribution and degradation" by Shang-Ze Li and colleagues describes the identification of a novel regulatory phosphorylation site (ser180) of the NF-kappaB sub-unit p65/RelA, target of the CLK2 kinase. CLK2 phosphorylation of p65 at ser180 by cytokine treatment or viral infection leads to increased p65 accumulation in the cytoplasm, and p65 proteasome-mediated degradation. Using knock-in phosphomimetic aspartic acid substitution of ser180 a decreased p65 activity is revealed compared to wt protein, following different stimuli, and also increased viral replication. On the contrary, substitution of ser180 with alanine, determining the inability of p65 to be phosphorylated by CLK2 at this residue, results in a "super-active" protein, capable of conferring resistance to viral infection in the knock-in cellular system. Moreover, CLK2 genetic ablation or its chemical inhibition promoted chronic inflammation in a mouse psoriasis model. Authors discussed CLK-2 mediated p65 phosphorylation at ser180 as an early NF-kappaB shutdown pathway, alternative to the canonical feed-back regulatory mechanism represented by the NF-kappaB-mediated synthesis of IKappaB-alpha occurring later after stimulation. The CLK-2 mediated phosphorylation of p65 and the subsequent inhibition of the activity of this transcription factor, represent important and novel findings, adding another important piece of information about how NF-kappaB, one of the most studied transcription factor, works. That being said, the manuscript, despite a large number of data gathered, lacks key experiments, and is misleading regarding some data interpretation. Moreover, an extensive revision for English language and grammar is required.

Specific comments:

1-Introduction section.

Introduction section is lacking the description of other important phosphorylation sites identified on p65 (by the way, naming this transcription factor as RelA is more appropriate). Among these, ser276, known to enhance NF-kappaB transcriptional activity, or ser468, negatively regulating NF-kappaB mediated transcription (for a review see "Posttranslational modifications of NF- κ B: another layer of regulation for NF- κ B signaling pathway" by Huang B. et al Cell Signal. 2010 September; 22(9): 1282–1290. doi:10.1016/j.cellsig.2010.03.017), the latter highly relevant in the context of this manuscript and used in fig. 6D and E panels. Moreover, the paper by Sacconi S. et al. "Degradation of Promoter-bound p65/RelA Is Essential for the Prompt Termination of the Nuclear Factor B Response" (J. Exp. Med. Volume 200, Number 1, July 5, 2004 107–113. doi: 10.1084/jem.20040196) must be acknowledged, due to fact that it describes p65/RelA degradation in the nucleus after activation, and in a DNA binding-dependent manner. In the same paper, if proteasome activity is blocked after an activating stimulus, NF-kappaB is not rapidly removed from target genes despite IKappaB synthesis, resulting in sustained transcription. Such paper describes a very similar mechanism of NF-kappaB "shutdown" and cannot be ignored. Finally, Lys 310 acetylation, required for full RelA transcriptional activity, should not be described without acknowledging the fact that it happens subsequently to the prior phosphorylation of RelA at ser276 and ser536 (Chen LF, et al. "NF-kappaB RelA phosphorylation regulates RelA acetylation". Mol Cell Biol. 2005 Sep;25(18):7966-75. doi: 10.1128/MCB.25.18.7966-7975.2005). What does "CLK2" stand for? Full name (Cdc2-like kinase 2) must be present, at least the first time CLK2 is described in the introduction section.

2- Fig.1

In the description of panel D, authors explain that they have used an ISRE reporter responding to SeV infection, to assess the impact of CLK2 on IRF-3, to rule out IRF-3 as a CLK2 target in IFN-beta promoter repression. However, authors did not provide evidence in the material and methods section about what specific ISRE sequence they have used in the reporter construct (indeed some commercially available ISRE reporter constructs also respond to IFN-I-induced ISGF3). In order for the authors to state that in fig. 1D they are dealing with IRF-3 mediated stimulation, they have to use the transfection of a plasmid constitutively expressing an active form of IRF-3, like IRF-3 5D (Lin R. et al. Essential Role of Interferon Regulatory Factor 3 in Direct Activation of RANTES Chemokine Transcription. Mol Cell Biol. 1999 Feb; 19(2): 959–966. doi: 10.1128/mcb.19.2.959), as a control, comparing the fold of activation obtained to those reached with SeV infection.

In Fig.1H error bars should be present in the graphs accounting for the three separate experiments, while the right western blot panel (showing TNF-alpha time course) should not present a GAPDH blot with such a poor separation between each band, clearly the result of excessive protein loading and/or overexposure.

3- Supplementary Fig.1.

Looking at panel 1D, it seems that CLK2 expression is actually negatively affected in p65 knock-out cells compared to wt cells, therefore the authors' conclusion, that CLK2 is not that much regulated at the level of transcription, is questionable; authors should provide a statistical analysis comparing each experimental time points of wt vs knock-out cells. Moreover, authors should also perform the same experiment using TNF-alpha as a stimulus with a proper related time course of treatment, to match with the experiments shown in fig. 1H.

4- Fig.2.

Panel B is not described in details and it is not clear, based on figure labelling, what does shown bands represent.

In panel E, particularly the right part involving MLFs, there is evidence for an increased accumulation of Nfkbia in the CLK2 -/- cells even in control untreated cells. This accumulation of Nfkbia matches with k310 acetylation and ser536 phosphorylation of p65, suggesting that, at least in these primary cells, CLK2 may represent a basal checkpoint acting to avoid a constitutive activation of p65 even in the absence of activating stimuli (e.g. LPS, TNF-alpha, IL-1 beta, or viral infection). It seems that in these cells there is also an increased accumulation of p65 compared to wt cells. Authors need to comment on all these data.

5- Supplementary Fig. 2.

Panel A is supposed to be positioned in the upper left part of the figure.

In the lower right part of panel F, (like for Fig.2 panel E, right part) there is evidence for an increased accumulation of Nfkbia in the CLK2 -/- cells even in control untreated cells. Even in this experiment, there is evidence for increased k310 acetylation and ser536 phosphorylation of p65 in control cells and increased accumulation of p65 compared to wt cells (the panel showing Ac-p65 Lys310 for the TNF-alpha time course must be repeated because bands are not clearly visible).

5-Fig.3

It is not clear what is the difference between Fig3C and supplementary Fig. 3E.

At line 169 of the result section related to Fig.3 ("CLK2 deficiency enhances virus-induced IFN-beta production and antiviral response") the word "plague" is actually "plaque", and need to be substituted throughout the manuscript.

6- Supplementary Fig. 3

Panel A is supposed to be positioned in the upper left part of the figure.

The rationale for the use of HSV-1 compared to VSV in in vivo and in vitro experiments should be explained, together with the different results obtained.

It is not clear what kind of supplemental information is provided with supplementary Fig.3E, that is not present in Fig.3C.

7-Fig.4

The conclusion, related to panel A, that "CLK2 inhibits virus-triggered NF-kappaB signaling at the p65 level" (lines 197-198 of the corresponding result paragraph) is correct, but does not rule out the possibility that there are also other upstream players in the signal transduction pathways affected by CLK2.

In panel B equal amount of Flag-CLK2 should be present in the lysate.

In panel C authors should also use a CLK2 inhibitor to conclude that CLK2 exerts its function with p65 depending on CLK2 kinase activity. This is because the lack of p65 binding to CLK2 K192R dead kinase, may also depend upon CLK2 conformational change due to the K to R mutation. Again, in panel C and F authors should comment about the faster migration of Flag-CLK2 dead kinase in the IP/WB experiment, compared to wt. This difference cannot be justified by K192R amino acid substitution.

In panel D, the western blotting experiment should also be repeated with an appropriate TNF-alpha time course.

In panel E, the graph should also report the luciferase fold of inductions relative to each uninfected control, to account for the basal activity of each kinase (wt and mutant) on the promoter compared to the SeV infection or TNF-alpha treatment.

In the western blot of panel F it is not clear why there is more p65 in the presence of wt CLK2 compared to CLK2 absence or the presence of the dead kinase. Authors should comment on this. In I and G panels, figures should be presented also showing cells with visible light, without fluorescence.

In panel H a quantification of the detected p65 bands should be presented.

8-Supplementary Fig.4

In panel A figures should be presented also showing cells with visible light, without fluorescence. In panel B, western blot should be repeated with a lower amount of loaded proteins (in GAPDH blot bands cannot be discriminated). Moreover, appropriate TNF-alpha time course should also be presented.

9-Fig.5

Panel distribution is confusing (what panel does the lower right graphic belongs to? Is it panel F? If this is the case, are those cells infected with SeV?

In panel C, figures should be presented also showing cells with visible light, without fluorescence. What is really missing, in this in depth phosphorylation site analysis involving p65, is that no attempt at all was made to identify a consensus amino acidic sequence on p65, target of the CLK2 kinase.

This is an important shortcoming of the manuscript and must be addressed in both, the result and the discussion paragraph.

10-Supplementary Fig.5

In panel D, figures should be presented also showing cells with visible light, without fluorescence.

11- Fig.6

Experiments shown in Panels D and E are very important for a crucial aspect of CLK2 mediated phosphorylation of p65, and that is p65 degradation and relocation of the protein from the nucleus to the cytoplasm. While MG132 experiments provide striking qualitative data demonstrating a role for phosphorylated ser180 in protein degradation, authors must calculate p65 wt, p65 K180A and p65 K180D half-life by performing a time course of cycloheximide (CHX) treatment, thus straightening their findings with solid quantitative data.

Moreover, as shown in CLK2 -/- cells a strong constitutive acetylation of p65 at K310 is detected, despite SeV infection or TNF-alpha treatment compared to its absence in wt cells and also an increased ser536 phosphorylation, known to precede K310 acetylation, compared to lower levels in wt cells.

This basal activation status of p65 in the absence of CLK2 and of any stimuli, suggest a different data interpretation (see comments below).

In the results paragraph describing Fig.6 and supplementary Fig.6, an important set of experiments is missing. Indeed no p65 DNA binding analysis was performed. Is p65 capable of binding kappaB sites when phosphorylated at ser 180 or when this residue is mutated to either aspartic acid or alanine? Particularly, is p65 capable of binding DNA kappaB sites in CLK2 knock-out cells, in the absence of activation stimuli, considering that, in this case, it is highly acetylated at K310, but at the same time it is supposed to be still bound to IkappaB-alpha? These are important questions that need to be answered by performing new experiments, also considering the work by Sacconi S. et al. (doi: 10.1084/jem.20040196), showing p65/RelA degradation in the nucleus after activation, and in a DNA binding-dependent manner.

At line 335 of the results paragraph " Ser-180 phosphorylation of p65 results in its degradation and nuclear export", ser-468 phosphorylation is described as an activation hallmark of p65, but in fact is mostly associated with a state of repression of p65 transcriptional activity (for a review see "Posttranslational modifications of NF-kB: another layer of regulation for NF-kB signaling pathway" by Huang B. et al Cell Signal. 2010 September; 22(9): 1282-1290.

doi:10.1016/j.cellsig.2010.03.017). Therefore, detection of ser 468 in panels D and E should be discussed accordingly.

Quantification is needed for the western blot results shown in panel C.

The following sentences " So far, these studies showed that phosphorylation of p65 at Ser-180 serves as a key switch in terminating active p65 through degradation and nuclear export. All of these experiments indicated that CLK2 acts as a brake in the early stage of NF-kappaB transcriptional activation, and it cooperates with IkappaB-alpha, which functions as a brake in the later stage, to limit the transcriptional process in the whole time of NF-kappaB signaling activation" present starting line 342 of the results paragraph " Ser-180 phosphorylation of p65 results in its degradation and nuclear export", should be reserved to the Discussion section. Moreover, these conclusions are only partially covering the large amount of data gathered, that also point to the CLK2 mediated phosphorylation of p65 at ser 180 as a homeostatic suppression of p65 activity before the signaling cascades of activating stimuli begin.

12- Supplementary Fig.6

In panel B it is not clear what n.s. (non specific) refers to. Is it compared to wt p65 or S180A mutant? It is important because it seems that there is an increased mRNA expression for the S180A mutant compared to wt or the S180D mutant.

In panel E it is not clear how this experiment was performed. Treatment of cells with MG132 results in the inhibition of p65 nuclear translocation and p65 mediated gene activation, due to the fact that IkappaB-alpha degradation is prevented (Lee, D.H. and Goldberg, A.L. 1998 "Proteasome inhibitors: valuable new tools for cell biologists". Trends Cell Biol. 8, 397-403 doi: 10.1016/s0962-8924(98)01346-4.). In this panel, though, authors show the nuclear accumulation of p65 S180D following TNF-alpha treatment in the presence of MG132. Authors should provide a plausible explanation for this considering that in this case IkappaB-alpha cannot be degraded. Moreover, IkappaB-alpha expression should be present in this western blot panel.

13-The following sentence "These data suggest that CLK2 represents a potential therapeutic target for inflammatory diseases", starting at line 374 of the results paragraph "CLK2 inhibitor TG003 exhibits power in vitro and in vivo", seems not formally correct. On the contrary, data suggest that any pathway inhibitor of CLK2 may represent a potential therapeutic target for inflammatory diseases, because CLK2 actually acts inhibiting prolonged inflammation.

14-Discussion

Discussion should provide a better data interpretation, also including discussion of results from new experiments related to p65 DNA binding.

Marco Sgarbanti PhD

Point-by-point response to reviewers' comments

We would like to thank the reviewers for their comments and insightful suggestions, which have greatly helped us to revise the manuscript and improve our study. Based on the reviewers' comments, we have performed additional experiments and clarified certain statements/experimental procedures in the revised manuscript. Following is our point-by-point response to the reviewers' comments.

Reviewer #1

The major finding of this manuscript is the identification of the NF- κ B p65 subunit as a target of CLK2. Phosphorylation of p65 at Ser-180 leads to nuclear export and degradation. This finding uncovers a new substrate for CLK2, a kinase previously found to modify splicing factors. Most importantly this manuscript discovers a new mechanism for negative feedback of p65-driven gene expression that is proposed to be independent of the well-known p65-dependent production of the I κ B α inhibitor. The authors use complimentary approaches to demonstrate that loss of CLK2, CLK2 kinase function, or p65 Ser-180 phosphorylation leads to a gain of interferon and inflammatory responses that have functional impacts on virus control and inflammatory pathologies, in cell culture and in mouse models. The parsing of main figure and supplementary figures is appropriate. That being said, there are a few holes in their data that should be addressed to bolster CLK2 as the direct p65 kinase that functions prior to and independently of I κ B α .

Major issues:

1. They indicate that CLK2 is an early terminator of NF- κ B signaling via p65 on Ser180 and independent of I κ B α - but there are some discrepancies with this interpretation.

a. Most assays are at 6, 12, or 24 hrs post-stim, well-after I κ B α is typically transcriptionally induced by p65. Is there clear data showing two waves of inhibition? Does I κ B α ever function in the absence of CLK2 or does CLK2 dominate?

Response:

We actually performed the TNF- α stimulation assays at 0.25, 0.5, 1, 3, and 6 hours, containing both the early and later stages of NF- κ B activation (Fig. 5g). The results showed that p65^{S180A}, which cannot be phosphorylated by CLK2, has higher transcriptional activity than p65^{WT}. Additionally, we did not observe two waves of inhibition. Our opinion is that the inhibition of nuclear p65 by CLK2 is persistent, and this event starts its progression once p65 shuttles into the nucleus. In other words, the occurrence of the event we defined occurs at the early stage of active NF- κ B, and the inhibition by resynthesis of I κ B α coexists with the inhibition by CLK2. Therefore, it does not have two waves of the inhibition.

Many results exhibit that I κ B α still function in *Clk2* knockout primary cells or *CLK2* knockout cell lines, such as Fig. 2e, Supplementary Fig. 2f and Fig. 6f.

For the question "does CLK2 dominate I κ B α ?", our opinion is that the role of CLK2 is to regulate p65 transcriptional activity at the post-translational modification (PTM) level. The reproduction of I κ B α is a downstream event of p65 phosphorylation by CLK2, similar to other PTM regulations, such as acetylation. Thus, CLK2 dominates the resynthesis of I κ B α but it does not dominate the function of I κ B α .

b. Does the phosphorylation change of p65 in response to CLK2 impact the ability of p65 to be bound by IκBα? If p65 is no longer recognized by IκBα in the absence of CLK2 modifications then that could lead to an accumulation of p65 in the nucleus and increased transcriptional activity. p65 S180A is a mutant that can't be phosphorylated downstream of CLK2 and has a prolonged residence in the nucleus (Fig 6), but how is the role of IκBα excluded? Do any of the phosphorylation site mutants impact targeting by IκBα?

Response:

We thank the reviewer for his/her insightful comments. We performed the exogenous coimmunoprecipitation assay to detect IκBα binding with p65^{WT} or p65^{S180D} (Supplementary Fig. 6h), and the result showed that p65^{S180D} has a higher affinity for IκBα. This indicates that phosphorylation at Ser180 increase the ability of p65 to be bound by IκBα.

In the absence of CLK2, p65 showed less affinity for IκBα, but it was still recognized by IκBα (Supplementary Fig. 6i). On the other hand, p65^{S180A/KI} cells were also used to confirm this result. The data showed that p65^{S180A/KI} decrease the interaction with IκBα (Fig. 6d), which is consistent with the result in CLK2^{-/-} cells. Taken together, these data suggest that CLK2 deficiency or p65 mutant that cannot be phosphorylated at Ser180 decrease the interaction between p65 and IκBα.

The results (Fig. 5g, 6b, and 6d) showed that p65^{S180A} has less affinity for IκBα and shows more nuclear localization and more powerful transcriptional activity. We did not exclude the role of IκBα and it still participates in the regulation of NF-κB signaling as a classic inhibitor.

Both of the experiments in p65^{S180A/KI} or p65^{S180D/KI} cells have suggested us that phosphorylation of p65 at Ser180 did impact targeting by IκBα (Fig. 6d and Supplementary Fig. 6h).

2. Is CLK2 the sole kinase and what activates it?

a. They identify multiple phosphorylation sites but there is never an in vitro analysis with purified CLK2 protein. As it stands now- CLK2 is certainly an upstream kinase- but given IP is performed in HEK293T - CLK2 might be acting via another kinase?

Does the CLK2 inhibitor TG003 block phosphorylation of Ser 180 on p65 or any other site?

Response:

What a great advice promotes the quality of the manuscript. We followed your advice and performed *in vitro* assays, including pull-down assay and kinase assay. Unfortunately, purification full-length CLK2 and CLK2^{K192R} active proteins through eukaryotic expression presents significant challenges for us. Therefore, we used the Flag-CLK2 and Flag-CLK2^{K192R} via eukaryotic expression, and His-p65 via prokaryotic expression. The pull-down assay showed that p65 only binds to CLK2, but not CLK2^{K192R} (Fig. 4f). Meanwhile, the *in vitro* kinase assay exhibited that CLK2 can expend more ATP with p65 (Fig. 4g).

The question raised by the reviewer is quite interesting and important for us to understand the mechanism in depth. Western blot analysis was performed with and without TG003, and the results showed that TG003 can block the phosphorylation of Ser-180 and promote the phosphorylation of Ser-536 and the acetylation of Lys-310 (Supplementary Fig. 6l).

Therefore, we believe the phosphorylation of p65 at Ser180 acting via CLK2 but not another kinase.

b. In Fig 6D, the SeV infected CLK2^{-/-} have lower levels of phospho Ser180 p65 overall, but it is not eliminated. What kinase drives this remaining phosphorylation? What does the asterisk/upper band indicate in the western blots?

Response:

The phosphorylation antibody was produced by Abclonal company. We consulted this question and according to the explanation of the technical support of the company, the phosphorylation of Ser-180 in the CLK2^{-/-} group can still be detected mainly due to

the purity of the antibody. The quality may not be perfect, and a small amount of total-p65 may also be detected. In summary, these unspecific bands do not affect the conclusion reflecting the detection of dynamic changes in the phosphorylation of Ser-180.

The asterisk/upper band indicate the unspecific band that can be detected by the antibody and the explanation was added in the relative figure legends.

c. This CLK2-mediated inhibition is only observed upon stimulation- so what is activating CLK2?

Response:

The question deserves attention and we did perform experiments to try to figure out what activates CLK2. The CLK2 is regulated obviously upon stimulation. Mass spectrometry and Co-IP experiments showed that CLK2 interacts with PLK1 exposed with the stimuli, suggesting that PLK1 may regulate CLK2 during infection. However, subsequent experiments cannot fully prove that CLK2 is a substrate of PLK1 or the relationship between PLK1 and CLK2 during infection. In the manuscript, we did not show the figures, but we mentioned in the discussion section that the activity of CLK2 is likely regulated during infection, possibly at the protein or kinase activity level.

3. There is a an overinterpretation that PML-associated p65 is more transcriptionally active. The PML colocalization and phase separation data is not well developed, being performed in the context of overexpression- it raises more questions than answers. Consider removing.

Response:

Thanks for your advice. The phase separation data truly raised more questions and we had removed it. However, the aggregation of p65^{S180A} and its co-localization with PML actually occurred and really interesting, we have decided to exhibit this phenomenon in the supplementary data.

4. A major flaw in the publication is the poor quality of writing. Professional editing is essential. There were numerous grammatical errors and improper word choices (e.g. 'we wonder CLK2 may be a pregnant target to treat inflammation-associated diseases and virus infection') that make it nearly impossible for the reader to work through. This did detract from my understanding of some of their summaries and interpretations.

Response:

We have revised the grammar throughout the manuscript and ask a few professors nearby to help us revise the manuscript. After the revision, the updated manuscript is now ready for exhibited.

Minor issues:

1. Fig 2B. The PCR genotyping data for WT vs KO mice seems swapped.

Response:

Thanks for this reminding. We did swap the WT vs KO mice PCR genotyping data and revised it in Fig. 2b.

2. Supp Fig 3 and Fig 5. have an odd arrangement of figures where the letters are not in alphabetical order.

Response:

Sorry for confusing the reviewer and now Supplementary Fig. 3 and Fig. 5 have been rearranged.

3. The font kerning for the sequence alignment of Fig 5B is off. A monospace font is needed.

Response:

Thanks for your suggestion. The monospace font has been adjusted.

4. Fig 6 Figure legend states data are presented as SEM- but that type of data is not in the figure.

Response:

Thanks. The legend of Fig. 6 has been revised.

5. Information on the strain and source of SeV, VSV, and HSV virus stocks is lacking.

Response:

SeV, VSV and VSV expressing GFP (VSV-GFP) were gifts from Hongbing Shu and Bo Zhong, which they have widely used in their studies and published a series of articles (Lei C. Q et al. Glycogen synthase kinase 3beta regulates IRF3 transcription factor-mediated antiviral response via activation of the kinase TBK1. *Immunity*. 878-89 (2010). doi: 10.1016/j.immuni.2010.11.021; Lian H, Zang R, Wei J, et al. The Zinc-Finger Protein ZCCHC3 Binds RNA and Facilitates Viral RNA Sensing and Activation of the RIG-I-like Receptors. *Immunity*. 2018;49(3):438-448.e5. doi: 10.1016/j.immuni.2018.08.014; Liuyu, T., Yu, K., Ye, L. et al. Induction of OTUD4 by viral infection promotes antiviral responses through deubiquitinating and stabilizing MAVS. *Cell Res* 29, 67–79 (2019). doi.org/10.1038/s41422-018-0107-6). The information was presented in the methods section.

Reviewer #2

The manuscript entitled “CLK2-mediated IkappaBalpha-independent early-termination of NF-kappaB activation by phosphorylating nuclear-p65 for cytoplasmic redistribution and degradation” by Shang-Ze Li and colleagues describes the

identification of a novel regulatory phosphorylation site (ser180) of the NF-kappaB sub-unit p65/RelA, target of the CLK2 kinase. CLK2 phosphorylation of p65 at ser180 by cytokine treatment or viral infection leads to increased p65 accumulation in the cytoplasm, and p65 proteasome-mediated degradation. Using knock-in phosphomimetic aspartic acid substitution of ser180 a decreased p65 activity is revealed compared to wt protein, following different stimuli, and also increased viral replication.

On the contrary, substitution of ser180 with alanine, determining the inability of p65 to be phosphorylated by CLK2 at this residue, results in a “super-active” protein, capable of conferring resistance to viral infection in the knock-in cellular system. Moreover, CLK2 genetic ablation or its chemical inhibition promoted chronic inflammation in a mouse psoriasis model.

Authors discussed CLK-2 mediated p65 phosphorylation at ser180 as an early NF-kappaB shutdown pathway, alternative to the canonical feed-back regulatory mechanism represented by the NF-kappaB-mediated synthesis of IKappaB-alpha occurring later after stimulation.

The CLK-2 mediated phosphorylation of p65 and the subsequent inhibition of the activity of this transcription factor, represent important and novel findings, adding another important piece of information about how NF-kappaB, one of the most studied transcription factor, works.

That being said, the manuscript, despite a large number of data gathered, lacks key experiments, and is misleading regarding some data interpretation.

Moreover, an extensive revision for English language and grammar is required.

Response:

The extensive revision for English language and grammar has been done. We have revised the grammar throughout the manuscript and ask a few professors nearby to help us revise the manuscript. After the revision, the updated manuscript is now ready for exhibited.

Specific comments:

1-Introduction section.

Introduction section is lacking the description of other important phosphorylation sites identified on p65 (by the way, naming this transcription factor as RelA is more appropriate). Among these, ser276, known to enhance NF-kappaB transcriptional activity, or ser468, negatively regulating NF-kappaB mediated transcription (for a review see “Posttranslational modifications of NF-κB: another layer of regulation for NF-κB signaling pathway” by Huang B. et al Cell Signal. 2010 September; 22(9): 1282–1290. doi:10.1016/j.cellsig.2010.03.017), the latter highly relevant in the context of this manuscript and used in fig. 6D and E panels. Moreover, the paper by Saccani S. et al. “Degradation of Promoter-bound p65/RelA Is Essential for the Prompt Termination of the Nuclear Factor B Response” (J. Exp. Med. Volume 200, Number 1, July 5, 2004 107–113. doi: 10.1084/jem.20040196) must be acknowledged, due to fact

that it describes p65/RelA degradation in the nucleus after activation, and in a DNA binding–dependent manner. In the same paper, if proteasome activity is blocked after an activating stimulus, NF-kappaB is not rapidly removed from target genes despite IkappaB synthesis, resulting in sustained transcription. Such paper describes a very similar mechanism of NF-kappaB “shutdown” and cannot be ignored. Finally, Lys 310 acetylation, required for full RelA transcriptional activity, should not be described without acknowledging the fact that it happens subsequently to the prior phosphorylation of RelA at ser276 and ser536 (Chen LF, et al. “NF-kappaB RelA phosphorylation regulates RelA acetylation”. Mol Cell Biol. 2005 Sep;25(18):7966-75. doi: 10.1128/MCB.25.18.7966-7975.2005).

What does “CLK2” stand for? Full name (Cdc2-like kinase 2) must be present, at least the first time CLK2 is described in the introduction section.

Response:

Regarding the name of p65, we believe your advice is valuable. The name as RelA is more official and we have updated it to RelA/p65 in the title and abstract sections. However, in consideration of the wildspread use of p65, we did not revise it in other sections.

Thank the reviewer for pointing these out for us to refine the writing of Introduction. The description of other important phosphorylation sites identified on p65 has been presented in the Introduction section. Among them, we have cited papers that focus on Ser276 and have also got acknowledge of the paper by Sacconi S. et al. Additionally, we have updated the description of Lys310 acetylation followed the reviewer’s advice. Meanwhile, a few more papers were cited in the introduction section (such as “Sgarbanti M, et al. IRF-1 is required for full NF-kappaB transcriptional activity at the human immunodeficiency virus type 1 long terminal repeat enhancer. J Virol. 2008;82(7):3632-3641. doi:10.1128/JVI.00599-07”; “Acchioni C, Remoli AL, Marsili G, et al. Alternate NF-κB-Independent Signaling Reactivation of Latent HIV-1 Provirus. J Virol. 2019;93(18): e00495-19. Published 2019 Aug 28. doi:10.1128/JVI.00495-19”). Here is the revised sentence “Over the last decades, significant progress has been made in understanding the nuclear import process and transcriptional activation modification of p65, such as phosphorylation of Ser-536 and Ser-276 is essential for the transactivation, phosphorylation of Ser-468 may negatively regulates NF-κB mediated transcription and acetylation of Lys-310 is required for full transcriptional activity of p65 subsequently to the prior phosphorylation of p65 at Ser-276 and Ser-536”.

The full name of CLK2 has been revised in the sentence of introduction section “In this study, we present an IκBα-independent nuclear NF-κB termination model at the transcriptional level through inhibitory phosphorylation by CDC like kinase 2 (CLK2).”

2- Fig.1

In the description of panel D, authors explain that they have used an ISRE reporter responding to SeV infection, to assess the impact of CLK2 on IRF-3, to rule out IRF-3 as a CLK2 target in IFN-beta promoter repression. However, authors did not provide

evidence in the material and methods section about what specific ISRE sequence they have used in the reporter construct (indeed some commercially available ISRE reporter constructs also respond to IFN-I-induced ISGF3). In order for the authors to state that in fig. 1D they are dealing with IRF-3 mediated stimulation, they have to use the transfection of a plasmid constitutively expressing an active form of IRF-3, like IRF-3 5D (Lin R. et al. Essential Role of Interferon Regulatory Factor 3 in Direct Activation of RANTES Chemokine Transcription. Mol Cell Biol. 1999 Feb; 19(2): 959–966. doi: 10.1128/mcb.19.2.959), as a control, comparing the fold of activation obtained to those reached with SeV infection.

Response:

Thanks for this professional reminder. The IFN-β and ISRE luciferase reporter plasmids and SeV, VSV and VSV expressing GFP (VSV-GFP) were gifts from Hongbing Shu and Bo Zhong (Wuhan University, Hubei, China), which is added in the methods section.

These luciferase reporter plasmids and virus were widely used in a lot of relatively and importantly published paper and the ISRE-Luc plasmid was tested by IRF-3 5D previously (Lei C. Q et al. Glycogen synthase kinase 3beta regulates IRF3 transcription factor-mediated antiviral response via activation of the kinase TBK1. Immunity. 878-89 (2010). doi: 10.1016/j.immuni.2010.11.021). Here is the specific sequence of ISRE “5'-AGGGAAAGTGAAACT-3'” and the information was presented in the methods section.

In Fig. 1H error bars should be present in the graphs accounting for the three separate experiments, while the right western blot panel (showing TNF-alpha time course) should not present a GAPDH blot with such a poor separation between each band, clearly the result of excessive protein loading and/or overexposure.

Response:

We do agree with the reviewer and have substituted the line charts with the data of three separate experiments (Fig. 1h).

3- Supplementary Fig. 1.

Looking at panel 1D, it seems that CLK2 expression is actually negatively affected in p65 knock-out cells compared to wt cells, therefore the authors' conclusion, that CLK2 is not that much regulated at the level of transcription, is questionable; authors should provide a statistical analysis comparing each experimental time points of wt vs knock-out cells. Moreover, authors should also perform the same experiment using

TNF-alpha as a stimulus with a proper related time course of treatment, to match with the experiments shown in fig. 1H.

Response:

With the experiment exposed with TNF- α and repeated experiment with SeV, we observed that p65 deficiency had a slight effect on the transcription of CLK2. However, compared to the transcription of other classic and specific genes induced by virus or cytokine-induced transcription, we still believe that the change in CLK2 mRNA is not obviously. (Supplementary Fig. 1d)

4- Fig.2.

Panel B is not described in details and it is not clear, based on figure labelling, what does shown bands represent.

In panel E, particularly the right part involving MLFs, there is evidence for an increased accumulation of Nfkbia in the CLK2 -/- cells even in control untreated cells. This accumulation of Nfkbia matches with k310 acetylation and ser536 phosphorylation of p65, suggesting that, at least in these primary cells, CLK2 may represent a basal checkpoint acting to avoid a constitutive activation of p65 even in the absence of activating stimuli (e.g. LPS, TNF-alpha, IL-1 beta, or viral infection). It seems that in these cells there is also an increased accumulation of p65 compared to wt cells. Authors need to comment on all these data.

Response:

We are sorry for not providing clear descriptions of the experiments. The band of WT represents the products that can only be amplified in CLK2^{WT} genome, and the band of KO represents the products that can only be amplified in the CLK2^{KO} genome. Additionally, the description of panel B has been revised in the figure legend.

Thanks for the insightful comment and comprehension of the reviewer and we deeply agree with it. In consideration of the dynamic balance in cell physiology, CLK2 deficiency may cause the slightly activation of NF- κ B signaling in the rest time and result in the increased accumulation of Nfkbia. The new description has been presented

in the result and discussion section “Interestingly, there was an increased accumulation of Nfkbia in *Clk2*^{-/-} cells, which is consistent with the increase in Lys-310 acetylation and Ser-536 phosphorylation of p65. It indicates that CLK2 may be a basal checkpoint to limit constitutional activation of NF-κB signaling, even in the absence of activating stimuli.” , “Taken together, all the results obtained from primary cells or cell lines indicated that the phosphorylation of p65 by CLK2 may be identified as a basal checkpoint, preventing a continuous activation of p65 and thus inhibiting the activation of NF-κB.” and “Furthermore, the CLK2 mediated phosphorylation of p65 at Ser-180 serves as a homeostatic suppression of p65 activity before the activation of the signaling cascade”.

5- Supplementary Fig. 2.

Panel A is supposed to be positioned in the upper left part of the figure.

In the lower right part of panel F, (like for Fig.2 panel E, right part) there is evidence for an increased accumulation of Nfkbia in the *CLK2*^{-/-} cells even in control untreated cells. Even in this experiment, there is evidence for increased k310 acetylation and ser536 phosphorylation of p65 in control cells and increased accumulation of p65 compared to wt cells (the panel showing Ac-p65 Lys310 for the TNF-alpha time course must be repeated because bands are not clearly visible).

Response:

Following the reviewer’s suggestion, we have rearranged Supplementary Fig. 2.

Thank the reviewer for pointing this out for us. The detection of Ac-p65 (Lys310) have been repeated with the same protein sample and the bands have been renewed (Supplementary Fig. 2f).

5-Fig.3

It is not clear what is the difference between Fig3C and supplementary Fig. 3E.

At line 169 of the result section related to Fig.3 (“CLK2 deficiency enhances virus-

induced IFN-beta production and antiviral response”) the word “plague” is actually “plaque”, and need to be substituted throughout the manuscript.

Response:

We are sorry for confusing the reviewer. We performed assays in 3 different types of primary cells, including bone marrow-derived macrophages (BMDMs), mice lung fibroblasts (MLFs) and bone marrow-derived dendritic cells (BMDCs). The data of BMDMs and MLFs were exhibited in Fig. 3 and the data of BMDCs was showed in Supplementary Fig. 3.

Thank the reviewer for point this out for us. We have corrected the plague to plaque throughout the manuscript.

6- Supplementary Fig. 3

Panel A is supposed to be positioned in the upper left part of the figure.

The rationale for the use of HSV-1 compared to VSV in in vivo and in vitro experiments should be explained, together with the different results obtained.

It is not clear what kind of supplemental information is provided with supplementary Fig.3E, that is not present in Fig.3C.

Response:

Following the reviewer’s suggestion, The Supplementary Fig. 3 has been rearranged.

Vesicular stomatitis virus (VSV) is an RNA virus and herpes simplex virus type 1 (HSV-1) is an DNA virus. Both of them can activate NF- κ B signaling but correspond to different receptor. All the stimulus we used were designed to verify that CLK2 is a broad-spectrum inhibitor of activated NF- κ B signaling. The explanation was presented in the sentence “To investigate the physiological role of Clk2 in viral infection *in vivo*, Clk2-deficient mice and their wild-type littermates were injected with Vesicular Stomatitis Virus (VSV) or Herpes Simplex Virus Type 1 (HSV-1) via the tail vein and intraperitoneally”.

We have confirmed the results through BMDMs and MLFs cells and these data are convincing. The experiments in BMDCs were mainly performed as a support of these known results, so the fluorescence microscopy of BMDCs was not detected. Considering it is supplementary data, we did not show it in the figure, but we still have strong evidence to proof the conclusion.

7-Fig.4

The conclusion, related to panel A, that “CLK2 inhibits virus-triggered NF-kappaB signaling at the p65 level” (lines 197-198 of the corresponding result paragraph) is correct, but does not rule out the possibility that there are also other upstream players in the signal transduction pathways affected by CLK2.

Response:

Thanks for the stringent description for this conclusion. It does not rule out the possibility that there are other upstream players in the signal transduction pathways

affected by CLK2. Therefore, we detected the location of CLK2 (Supplementary Fig. 4a). Combined with the luciferase and nuclear location results, we draw the conclusion that CLK2 function with nuclear p65 to regulate the activation of the NF- κ B signaling.

In panel B equal amount of Flag-CLK2 should be present in the lysate.

In panel C authors should also use a CLK2 inhibitor to conclude that CLK2 exerts its function with p65 depending on CLK2 kinase activity. This is because the lack of p65 binding to CLK2 K192R dead kinase, may also depend upon CLK2 conformational change due to the K to R mutation.

Response:

Thanks for your rigorous manner, the expression of Flag-CLK2 did have discrepancy and it may be caused by plasmid transfection. However, this discrepancy did not affect our conclusion and we did not replace it.

We have followed the reviewer’s advice. The assay was performed and the result showed that the interaction between p65 and CLK2 is impaired in the presence of the CLK2 inhibitor-TG003 (Fig. 4d).

By the way, The CLK2^{K192R} was widely used in many other researches (Nayler, O et al. “The cellular localization of the murine serine/arginine-rich protein kinase CLK2 is regulated by serine 141 autophosphorylation.” The Journal of biological chemistry vol. 273,51 (1998): 34341-8. doi:10.1074/jbc.273.51.34341; Rodgers, Joseph T et al. “Cdc2-like kinase 2 is an insulin-regulated suppressor of hepatic gluconeogenesis.” Cell metabolism vol. 11,1 (2010): 23-34. doi:10.1016/j.cmet.2009.11.006). And the strategy of frequently replacing Lysine (Lys) in kinase activity mutations was normally used in many other researches (Hanafusa H, Kedashiro S, Tezuka M, et al. PLK1-dependent activation of LRRK1 regulates spindle orientation by phosphorylating CDK5RAP2. Nat Cell Biol. 2015;17(8):1024-1035. doi:10.1038/ncb3204; Zhao P, Wong KI, Sun X, et al. TBK1 at the Crossroads of Inflammation and Energy Homeostasis in Adipose Tissue. Cell. 2018;172(4):731-743.e12. doi:10.1016/j.cell.2018.01.007). So we choose the CLK2^{K192R} to investigate the function of kinase activity.

Again, in panel C and F authors should comment about the faster migration of Flag-CLK2 dead kinase in the IP/WB experiment, compared to wt. This difference cannot be justified by K192R amino acid substitution.

Response:

It may be caused by the site mutation and result in the change of PTM, such as autophosphorylation. The migration of bands can be observed normally in many other kinase dead mutation. Not only in our research, but also many others research (Zhao P, Wong KI, Sun X, et al. TBK1 at the Crossroads of Inflammation and Energy Homeostasis in Adipose Tissue. *Cell*. 2018;172(4):731-743.e12. doi:10.1016/j.cell.2018.01.007; Li SZ, Shu QP, Song Y, et al. Phosphorylation of MAVS/VISA by Nemo-like kinase (NLK) for degradation regulates the antiviral innate immune response. *Nat Commun*. 2019;10(1):3233. Published 2019 Jul 19. doi:10.1038/s41467-019-11258-x).

In panel D, the western blotting experiment should also be repeated with an appropriate TNF-alpha time course.

Response:

Following the reviewer's suggestion, the experiment has been repeated exposed with TNF- α , and the result showed that p65 has more affinity to CLK2 under stimulation.

In panel E, the graph should also report the luciferase fold of inductions relative to each uninfected control, to account for the basal activity of each kinase (wt and mutant) on the promoter compared to the SeV infection or TNF-alpha treatment. In the western blot of panel F it is not clear why there is more p65 in the presence of wt CLK2 compared to CLK2 absence or the presence of the dead kinase. Authors should comment on this.

Response:

Sorry for confusing the reviewer. About panel E (updated as Supplementary Fig. 4b), the uninfected control group was been showed which was painted green.

The difference in panel F (rearranged as Supplementary Fig. 4c) as most likely caused by transfection or western blot system error. It was ignored and actually made some misunderstands. However, it was not a quantitative experiment and did not affect us drawing the conclusion. We have repeated the experiment multiple times and in other

cases, it is not observed the same increased expression. Two additional repetitions are shown below. We have also rerun the same protein sample and substituted the improper graph. Finally, thanks for careful observation and reminding.

In I and G panels, figures should be presented also showing cells with visible light, without fluorescence.

Response:

We could understand what your concern. Usually, visible light is not showed in the figure, so we did not shoot the cells with visible (Tanaka, T., Grusby, M. & Kaisho, T. PDLIM2-mediated termination of transcription factor NF- κ B activation by intranuclear sequestration and degradation of the p65 subunit. *Nat Immunol* 8, 584–591 (2007). <https://doi.org/10.1038/ni1464>; Wang, S., Lin, Y., Yuan, X. et al. REV-ERB α integrates colon clock with experimental colitis through regulation of NF- κ B/NLRP3 axis. *Nat Commun* 9, 4246 (2018). <https://doi.org/10.1038/s41467-018-06568-5>). Instead, DAPI was used to reflect the number and condition of cells as a merged graph, and we did detect it. It is shown below, but in order to clearly show the change of p65, we did not include it in the manuscript.

In panel H a quantification of the detected p65 bands should be presented.

Response:

Following the reviewer's suggestion, we have presented the line chart and updated it in Fig. 4i.

8-Supplementary Fig.4

In panel A figures should be presented also showing cells with visible light, without fluorescence.

Response:

Thanks. It is a similar question to 7-Fig. 4h and 4j, and about the question, it had been explained.

In panel B, western blot should be repeated with a lower amount of loaded proteins (in GAPDH blot bands cannot be discriminated). Moreover, appropriate TNF-alpha time course should also be presented.

Response:

We agree with the reviewer that western blot need to be revised and the updated graphs (Supplementary Fig. 4d). The results showed that CLK2 deficiency does not affect p65 stability under the stimulation.

9-Fig.5

Panel distribution is confusing (what panel does the lower right graphic belongs to? Is it panel F? If this is the case, are those cells infected with SeV?)

Response:

We feel so sorry for confusing the reviewer and the graph has been rearranged.

The lower right graphic belongs to panel F, and those cells were same sample that treated with SeV. Now it has been rearranged as Fig. 5e.

In panel C, figures should be presented also showing cells with visible light, without fluorescence.

Response:

It is a similar question to 7-Fig. 4h and 4j, and about the question, we have been explained previously.

What is really missing, in this in depth phosphorylation site analysis involving p65, is that no attempt at all was made to identify a consensus amino acidic sequence on p65, target of the CLK2 kinase.

This is an important shortcoming of the manuscript and must be addressed in both, the result and the discussion paragraph.

Response:

Thanks for giving the valuable suggestions. We really appreciate it and have attempted to determine if there is a consensus amino acid sequence that is targeted by CLK2. Unfortunately, the only confirmed phosphorylated site of the substrate targeted by CLK2 is PTP1B, and there has not enough data to figure out the specific motif. Finally, we have updated it in the discussion section “Following the conserved sites analysis of p65 180 and 316 residues across species Fruit fly to Human, we tried to identify the consensus amino acid sequence on p65 targeted by CLK2. The PROMALS3D online tool was used for this investigation and yielded no significant result. Given that Ser-50 of PTP1B is the sole site that has been determined targeted by CLK2, and no additional substrates can be concurrently analysed, the determination of this target remains challenging. However, further investigation holds promise for its eventual elucidation.”.

10-Supplementary Fig.5

In panel D, figures should be presented also showing cells with visible light, without fluorescence.

Response:

It is a similar question to 7-Fig. 4h and 4j, and about the question, we have been explained previously.

11- Fig.6

Experiments shown in Panels D and E are very important for a crucial aspect of CLK2 mediated phosphorylation of p65, and that is p65 degradation and relocation of the protein from the nucleus to the cytoplasm. While MG132 experiments provide striking qualitative data demonstrating a role for phosphorylated ser180 in protein

degradation, authors must calculate p65 wt, p65 K180A and p65 K180D half-life by performing a time course of cycloheximide (CHX) treatment, thus straightening their findings with solid quantitative data.

Response:

Following the reviewer’s suggestion, we performed CHX assays and the results showed that p65^{S180A/KI} is more stable than p65^{WT}. Additionally, p65^{S180D/KI} has a much shorter half-life compared to p65^{WT} (Supplementary Fig. 6g). We also update the result section “Moreover, the nuclear p65^{S180D} had showed higher ubiquitination level than wild-type p65 (Supplementary Fig. 6f), and the cycloheximide (CHX) treatment assay also exhibited that p65^{S180D} has a shorter half-life than p65^{WT} (Supplementary Fig. 6g).”

G

Moreover, as shown in CLK2 -/- cells a strong constitutive acetylation of p65 at K310 is detected, despite SeV infection or TNF-alpha treatment compared to its absence in wt cells and also an increased ser536 phosphorylation, known to precede K310 acetylation, compared to lower levels in wt cells.

This basal activation status of p65 in the absence of CLK2 and of any stimuli, suggest a different data interpretation (see comments below).

In the results paragraph describing Fig.6 and supplementary Fig.6, an important set of experiments is missing. Indeed no p65 DNA binding analysis was performed. Is p65 capable of binding kappaB sites when phosphorylated at ser 180 or when this residue is mutated to either aspartic acid or alanine? Particularly, is p65 capable of binding DNA kappaB sites in CLK2 knock-out cells, in the absence of activation stimuli, considering that, in this case, it is highly acetylated at K310, but at the same time it is supposed to be still bound to IkappaB-alpha? These are important questions that need to be answered by performing new experiments, also considering the work by Saccani S. et al. (doi: 10.1084/jem.20040196), showing p65/RelA degradation in the nucleus after activation, and in a DNA binding-dependent manner.

Response:

It is really a good question and we have followed the reviewer to perform the DNA-binding analysis with the knock-in cells. The result of ChIP-qPCR showed that p65^{S180A/KI} has more affinity for κB sites than p65^{WT} under SeV stimulation, while p65^{S180D/KI} showed almost no binding with DNA (Fig. 6e).

It is similar to the result of p65^{S180A/KI}, in the absence of CLK2, wild-type p65 shows more powerful DNA-binding ability (Supplementary Fig. 6j). And the manuscript has been updated as “As known, interaction with IκBα is usually relevant to the DNA-binding activity, so we next sought for if p65^{S180D/KI} affects p65 binding with DNA. CHIP-qPCR assays showed that p65^{S180D/KI} has nearly lost the ability of DNA-binding, on the contrary, p65^{S180A/KI} exhibits the enhanced DNA-binding activity (Fig. 6e). In addition, p65 in CLK2^{-/-} cells also showed increase capability of DNA-binding (Supplementary Fig. 6j).”

J

And the question about why p65 supposed to be bound with IκBα has high level of Lys310 acetylation, we need to say that is not contradictory. According to the paper “Acetylation of RelA at discrete sites regulates distinct nuclear functions of NF-κB” (doi.org/10.1093/emboj/cdf660), acetylation at K221 enhances the DNA binding of NF-κB and, together with acetylation at K218, impairs its association with IκBα. Acetylation of K310 is required for full transcriptional activity of NF-κB, but does not affect DNA binding or IκBα assembly. And every molecule has its dynamic balance, not all of the p65 binds with IκBα, it just takes up a portion of it. For another example, p65/p50 dimer binding with IκBα dynamically shuttles between the nucleus and cytoplasm and basic transcription mediated by p65 is continuously proceeding, even in the rest time without stimulus, which maintains the dynamic balance of cell physiology (Huang TT, et al. A nuclear export signal in the N-terminal regulatory domain of IκappaBalpha controls cytoplasmic localization of inactive NF-kappaB/IkappaBalpha complexes. Proc Natl Acad Sci U S A. 2000;97(3):1014-1019. doi:10.1073/pnas.97.3.1014; Johnson C, Van Antwerp D, Hope TJ. An N-terminal nuclear export signal is required for the nucleocytoplasmic shuttling of IκappaBalpha. EMBO J. 1999;18(23):6682-6693. doi:10.1093/emboj/18.23.6682). And IκBα keeps

degradation and resynthesis in the rest time. Taken together, we can conclude that deficiency of CLK2 may result in accumulation of acetylation of a portion of p65. It enhanced the basic transcriptional activity of p65 and it is not related to binding with I κ B α .

Considering the good work by Sacconi S. et al. (doi: 10.1084/jem.20040196), we can observe that the mutant p65 did not eliminate the polyubiquitination (Fig. 2c) and the mRNA of TNF- α was not up-regulated by TNF- α in I κ B α ^{-/-} cells treated with β lactone (Fig. 3b). It suggests that degradation of promoter-bound p65 takes up a portion of p65 degradation, and there still have other mechanism. It is not contradictory.

At line 335 of the results paragraph “ Ser-180 phosphorylation of p65 results in its degradation and nuclear export”, ser-468 phosphorylation is described as an activation hallmark of p65, but in fact is mostly associated with a state of repression of p65 transcriptional activity (for a review see “Posttranslational modifications of NF- κ B: another layer of regulation for NF- κ B signaling pathway” by Huang B. et al Cell Signal. 2010 September; 22(9): 1282–1290. doi:10.1016/j.cellsig.2010.03.017). Therefore, detection of ser 468 in panels D and E should be discussed accordingly. Quantification is needed for the western blot results shown in panel C.

Response:

Thanks for figure out our mistakes and it has been revised throughout the paragraph. And about the result of increased Ser-468 phosphorylation in CLK2^{-/-} cells, here is the discussion. Ser-468 phosphorylation is an activation hallmark of p65 and the phosphorylation occurs in the nucleus (DOI 10.1074/jbc.M508045200) which links to the negatively regulation of p65 by degradation through proteasome (doi: 10.1038/embor.2009.10; doi: 10.1101/gad.1748409). According to the result in the paper (<https://doi.org/10.1096/fj.05-3736fje>), we can also find that Ser468 just has a slight effect in transactivation and does not alter the p65 translocation. Moreover, the review “Posttranslational modifications of NF- κ B: another layer of regulation for NF- κ B signaling pathway” also said “Surprisingly, phosphorylation of S468 by IKK ϵ in T cells enhances the transcriptional activation of NF- κ B in response to T cell co-stimulation (DOI 10.1074/jbc.M508045200). These different outcomes suggest that phosphorylation of S468 regulates the transcriptional activity of NF- κ B in a context-dependent manner, but the mechanism underlying this requires further investigation”. Actually, in normal experiments we usually cannot find a significant change in p65 protein levels under the stimulation. Previous research has not clearly clarified the role of Ser-468 phosphorylation, but it is a consensus that Ser-468 phosphorylation reflects the activation of p65. Combined with our results, it is obvious that CLK2 plays a key role in supervising the posttranslational modifications of p65. The nuclear accumulation of p65 caused by CLK2 deficiency may result in the accumulation of phosphorylated p65 at Ser-468. The result was revised as “The Ser-536 phosphorylation and Lys-310 acetylation is required for p65 full transcriptional activity, and Ser-468 phosphorylation mainly negatively regulates the transcriptional activity and also can be treated as a hallmark of activation of p65”.

The quantification of panel C has been shown in Fig. 6.

The following sentences “So far, these studies showed that phosphorylation of p65 at Ser-180 serves as a key switch in terminating active p65 through degradation and nuclear export. All of these experiments indicated that CLK2 acts as a brake in the early stage of NF-kappaB transcriptional activation, and it cooperates with IkappaB-alpha, which functions as a brake in the later stage, to limit the transcriptional process in the whole time of NF-kappaB signaling activation” present starting line 342 of the results paragraph “ Ser-180 phosphorylation of p65 results in its degradation and nuclear export”, should be reserved to the Discussion section. Moreover, these conclusions are only partially covering the large amount of data gathered, that also point to the CLK2 mediated phosphorylation of p65 at ser 180 as a homeostatic suppression of p65 activity before the signaling cascades of activating stimuli begin.

Response:

As suggested, the mentioned sentence was moved to the Discussion section.

Also, we agree the statement that “Furthermore, the CLK2 mediated phosphorylation of p65 at Ser-180 serves as a homeostatic suppression of p65 activity before the activation of the signaling cascade” and put it in the Discussion section following above sentence.

12- Supplementary Fig.6

In panel B it is not clear what n.s. (non specific) refers to. Is it compared to wt p65 or S180A mutant? It is important because it seems that there is an increased mRNA expression for the S180A mutant compared to wt or the S180D mutant.

Response:

Sorry for confusing the reviewer, the n.s. of p65^{S180D/KI} is compared to the p65^{WT}. And there is no significant difference between p65 and p65^{S180A/KI} through statistical analysis.

In panel E it is not clear how this experiment was performed. Treatment of cells with MG132 results in the inhibition of p65 nuclear translocation and p65 mediated gene activation, due to the fact that IkappaB-alpha degradation is prevented (Lee, D.H. and

Goldberg, A.L. 1998 “Proteasome inhibitors: valuable new tools for cell biologists”. *Trends Cell Biol.* 8, 397–403 doi: 10.1016/s0962-8924(98)01346-4.). In this panel, though, authors show the nuclear accumulation of p65 S180D following TNF-alpha treatment in the presence of MG132. Authors should provide a plausible explanation for this considering that in this case IκappaB-alpha cannot be degraded. Moreover, IκappaB-alpha expression should be present in this western blot panel.

Response:

The p65/p50 dimer binding with IκBα dynamically shuttles between the nucleus and cytoplasm. And basic transcription, mediated by p65, continuously proceeds even during rest period without stimulation. This maintains the dynamic balance of cell physiology. The experiment was used p65^{S180D/KI} HEK293T cells treated with TNF-α with or without MG132 and then fractionated the cytoplasm and nucleus to detect p65.

Regarding the inhibition of IκBα, we have a statement previously that even during the rest period, physiological progression continuously proceeds at a relatively low level. Not all of the p65 binds with IκBα, and a portion of p65 can shuttle from the cytoplasm to the nucleus to maintain the dynamic balance. When exposed with MG132, the result indicated that p65^{S180D} is largely accumulated in the nucleus. Meanwhile, the cytoplasmic IκBα, may be similar to the free resynthesized IκBα, shuttles from the cytoplasm to the nucleus. The updated Supplementary Fig. 6e is presented below.

13-The following sentence “These data suggest that CLK2 represents a potential therapeutic target for inflammatory diseases”, starting at line 374 of the results paragraph “CLK2 inhibitor TG003 exhibits power in vitro and in vivo”, seems not formally correct. On the contrary, data suggest that any pathway inhibitor of CLK2 may represent a potential therapeutic target for inflammatory diseases, because CLK2 actually acts inhibiting prolonged inflammation.

Response:

We agree with the reviewer and this sentence has been revised as following “These data suggest that CLK2 represents a potential therapeutic target for inflammatory diseases and TG003 may be developed as a cure of virus-infection.”

14-Discussion

Discussion should provide a better data interpretation, also including discussion of results from new experiments related to p65 DNA binding.

Response:

Discussion related to p65 DNA-binding has been added in the result and discussion section “As known, interaction with I κ B α is usually relevant to the DNA-binding activity, so we next sought for if p65^{S180D/KI} affects p65 binding with DNA. ChIP-qPCR assays showed that p65^{S180D/KI} has nearly lost the ability of DNA-binding, on the contrary, p65^{S180A/KI} exhibits the enhanced DNA-binding activity (Fig. 6e). In addition, p65 in CLK2^{-/-} cells also showed increase capability of DNA-binding (Supplementary Fig. 6j). In summary, phosphorylation of p65 by CLK2 at Ser-180 decreases the DNA-binding activity of p65 and enhance its interaction with CRM1 and I κ B α , resulting in the export of p65 from the nucleus.” and “Additionally, the p65^{S180A} and p65 in CLK2^{-/-} cells showed less affinity for CRM1 and I κ B α and a stronger DNA-binding activity, further demonstrating how Ser-180 phosphorylation affects the transcription.”.

REVIEWER COMMENTS

Reviewer #1 (Remarks to the Author):

Summary

The authors have provided a thorough rebuttal. They note numerous figure modifications with the addition of new data and removal and rearrangement of previous data, and text clarifications that have substantially improved the manuscript. The authors have defined a role for CLK2 in the impairment of NF- κ B activation and inflammation that impacts virus replication, and further improvements were made to Figs 1-3 in response to Reviewer 2. However, I am not satisfied with the mechanistic underpinnings of the impact of CLK2 on RelA/p65. Without confirmation that CLK2 directly phosphorylates RelA/p65 and that this inhibition is truly independent of I κ B α , the novelty of this publication is substantially reduced. The data provided does not strongly lock down these two claims.

Major concerns

1. Weak evidence that RelA/p65 Ser180 is a direct substrate of CLK2.

-For IP interactions and phos-tag and ADP glo kinase assays in Fig 4 and Supp Fig4: It would be ideal to perform with purified proteins. As it stands now this interaction could be part of a complex and not direct. They describe Fig 4f as an in vitro pulldown that I suppose is linked to the 'protein purification and pull-down' in Methods. This describes a purified His-p65 with FLAG- enriched lysates of transfected 293T cells. Is there a Coomassie stained gel to demonstrate the purity of the FLAG enriched lysate used for the FLAG pull-down?

-The authors respond in the rebuttal that the CLK2K192R mutant is widely used in the field. Can they verify that the CLK2 K192R mutant is kinase dead in this system using canonical substrates? It is also notable that CLK2 undergoes autophosphorylation on Ser141, mutation changes its subnuclear localization (PMID 9852100). Does the K192R prevent this autophosphorylation? One potential explanation is that the autophosphorylated form of CLK2 is required to interact with p65 (whether due to TG003 or the K192R mutant), but does not reflect the requirement for its kinase activity.

- TG003 is not specific for CLK2, this drug also inhibits other CLKs. This is important since CLK1,2,4 each regulate other substrates including splicing factors (PMID 33846420). They need to demonstrate that this is not impacting other CLKs or that other CLK2 substrates aren't playing a role. Why not show that the impact of TG003 is only observed in CLK2+, but not CLK2-/- cells.

-The guidelines and other publications that use the ADP-glo kinase assay report activity based on the change in luminescence with concentration of the protein/inhibitors present. Here, they apply it as a single timepoint/readout luminescent reading- they do not report kinase activity differences in CLK compared to the CLK2 K192R mutant over a range of protein/substrate concentrations. Given that CLK2 K192R is expressed at lower levels than CLK2 in Fig 4C- this decrease in luminescence might just represent a decrease in CLK2 mutant expression levels. They don't show assay results for FLAG-CLK2K192R in the absence of His-p65. What is the source of the samples examined (lysates or purified components)? The input proteins need to be shown to verify equal inputs. Minimally, a purified CLK2 kinase enzyme system is commercially available (Promega) and could be used with their purified p65 to verify p65 is a substrate for CLK2.

-The rationale for pursuing S180 but not S316 does not seem logical since the alignments show similar patterns of conservation across species for both (Fig 5B).

-The analysis of the p65 S180 mutant clearly shows that S180A is more active than S180D, but the stronger experiment is to show that both retain this profile with or without CLK2 expression, unlike WT p65 (Fig 5E).

-There was a notable difference in the expression level of HA-S180D in Fig 5 due to a decrease in stability; that is the simple explanation for loss of NF- κ B activation in Fig 4 functional assays. Why

aren't these assays shown in the presence and absence of CLK2 to demonstrate that only the WT p65 half-life/instability (Supp Fig 6G) is different between CLK2+ and CLK2-/- cells (S180A and S180D would not show any dependence on CLK2 correct, correct)?

-I remain in doubt about the findings using the Ser-180 p-p65 antibody (Fig 6F). The phospho-specific Ser-180 antibody that they had produced shows a non-specific band and evidently cross reacts with total p65, As controls, did they profile WT p65, S180A and S180D mutants to show this only recognizes the WT p65. Did they test p65-/- cells?

-Also there is inconsistency between the CLK2+/+ vs -/- cells in Fig6D at basal (0hrs) conditions. The p-p65 Ser180 is the same at 0 hr in the SeV blot on left but higher in CLK2+/+ cells compared to CLK2-/- cells at 0 hrs in the right TNFa blot.

2. I am still not convinced that CLK2 function is independent of IkBa.

"Interestingly, there was an increased accumulation of Nfkb1a in Clk2-/- cells, which is consistent with the increase in Lys-310 acetylation and Ser-536 phosphorylation of p65. It indicates that CLK2 may be a basal checkpoint to limit constitutional activation of NF-κB signaling, even in the absence of activating stimuli."

-I remain unclear what they mean by 'early stage termination.' It seems that the impact of CLK2 on p65 and Nfkb1a is most apparent in the context of induction than basal conditions. IkBa is expressed and plays a role in basal and induced conditions over a 6 hr timecourse.

-They state on lines 81-82 that 'we report an IkBa-independent nuclear NF-κB termination model at the transcriptional level through inhibitory phosphorylation by CLK2' Given that they have greater interactions of p65 with IkBa in the presence of CLK2 (without or with TNF) in Supp Fig 6I, doesn't this again indicate that IkBa can be playing a role in CLK2's inhibition of p65? Can they demonstrate that CLK2 suppresses p65 even in the absence of IkBa?

-I don't understand their statement that IkBa functions in CLK2 KO cells- the data they cite just shows Nfkb1a is resynthesized by western. Do they introduce IkBa into CLK2-/- cells and show it is still inhibitory?

-They find that S180D (which does not activate NF-κB and has a higher turnover) has a higher affinity for Nfkb1a/IkBa (Supp Fig 6H), but S180A and WT in CLK2-/- cells has a decreased interaction with CRM1 and Ikba (Fig 6C & 6D). This data demonstrates that the phosphorylation on S180 induced by CLK2 leads to a higher affinity for IkBa that would export and lock p65 out of nucleus; in turn this lead to the loss of NF-κB post-translational modifications and functions in the nucleus. Therefore, the impact of CLK2 on p65 Ser180 phosphorylation may be upstream of IkBa interactions, but ultimately is determined by IkBa export. It is not independent of Ikba- just not dependent on newly synthesized IkBa in late stage termination. This does not fit with their conclusions and title.

-Lines 368-370 described showing p65 S180A and WT interactions with CRM1 and IkBa in CLK2-/- cells. Fig 6D only shows 293T cells and Fig.6i only shows WT p65 in both CLK2+/+ and CLK2-/- cells.

Minor comments.

1. Professional editing is required even with revisions. Numerous issues remain and some are newly introduced. 'plague' is still used where 'plaque' is appropriate.
2. The IFA figures with brightfield added have become very small (Fig3, 5). The IFA for the paired flow cytometry data could be moved to supplemental or the other bar graphs in the figures could be shortened to make space.
3. I previously noted that the font kerning for the sequence alignment of Fig 5B is off and that a monospace font is needed. The authors claim to have adjusted this- but the text is still not aligned properly.
4. There is a lack of discussion regarding its canonical substrates such as splicing factors vs this

noncanonical functions of CLK2 to target p65. How does SeV, TNFa or other stimulations that activated CLK2 tie into its impacts on splicing? When CLK2 is activated by infection to target p65, what is the impact on canonical substrates?

5. Typo in Supp Fig 4c legend line 1057: they state 'The upper shifted band represents phosphorylated CLK2 protein.' This is actually p65, correct?

6. With regard to Fig 6D legend, they state that "The asterisk/upper band indicate the unspecific band that can be detected by the antibody and the explanation was added in the relative figure legends." This has not been corrected.

Reviewer #2 (Remarks to the Author):

Authors fixed all relevant issue raised, by performing new experiments as requested, and by modifying the manuscript, as suggested. I also encourage authors to make a further effort to improve manuscript written English (at pag 5 line 55, "rheumatoid arthritis", instead of "rheumatic arthritis"; at pag.6 line 89 "CLK2-deficient mice were observed to have higher levels", instead of "CLK2-deficient mice produce were observed to have higher levels"; at pag 22 line 444 "The transcription factor p65 is critical for cytokine-induced production of inflammation genes", instead of "The transcription factors p65 is critically for cytokine-induced production of inflammation genes"; at page 24 line 499 " use of site-specific phosphorylation antibody", instead of "use of stie-specific phosphorylation antibody", and so on).

Moreover, at pag. 21 line 440, I suggest to rephrase the sentence "... and TG003 may be developed as a cure of virus-infection" as " ...and CLK2 inhibition may be exploited as a cure of virus-infection", due to the fact that TG003 inhibitor may not be suitable for further chemical optimization, to serve as an effective and safe antiviral.

Marco Sgarbanti

Point-by-point response to reviewers' comments

We are extremely thankful for the reviewers' constructive feedback and valuable suggestions, which have greatly contributed to the enhancement of our manuscript and study. In response to the reviewers' comments, we have conducted additional experiments and made necessary clarifications to certain statements and experimental procedures in the revised manuscript. Here is our point-by-point response to the reviewers' comments.

Reviewer #1 (Remarks to the Author):

Summary

The authors have provided a thorough rebuttal. They note numerous figure modifications with the addition of new data and removal and rearrangement of previous data, and text clarifications that have substantially improved the manuscript. The authors have defined a role for CLK2 in the impairment of NF- κ B activation and inflammation that impacts virus replication, and further improvements were made to Figs 1-3 in response to Reviewer 2. However, I am not satisfied with the mechanistic underpinnings of the impact of CLK2 on RelA/p65. Without confirmation that CLK2 directly phosphorylates RelA/p65 and that this inhibition is truly independent of I κ B α , the novelty of this publication is substantially reduced. The data provided does not strongly lock down these two claims.

Major concerns

1. Weak evidence that RelA/p65 Ser180 is a direct substrate of CLK2.

-For IP interactions and phos-tag and ADP glo kinase assays in Fig 4 and Supp Fig4: It would be ideal to perform with purified proteins. As it stands now this interaction could be part of a complex and not direct. They describe Fig 4f as an in vitro pulldown that I suppose is linked to the protein purification and pull-down in Methods. This describes a purified His-p65 with FLAG- enriched lysates of transfected 293T cells. Is there a Coomassie stained gel to demonstrate the purity of the FLAG enriched lysate used for the FLAG pull-down?

Response:

It is really a valuable advice and we also agree that it would be ideal to use purified proteins. It may be complex and not show the interaction directly. We successfully purified His-p65 in prokaryotic cells, but were unable to purify full-length CLK2 and CLK2^{K192R} in prokaryotic cells. Instead, we chose to enrich them in transfected HEK293T cells. And the method is linked to the “protein purification and pull-down” in Methods section. Additionally, we actually ran a Coomassie stained gel, but it was inadvertently not provided. Here is the new version of Fig. 4f which is presented below.

-The authors respond in the rebuttal that the CLK2K192R mutant is widely used in the field. Can they verify that the CLK2 K192R mutant is kinase dead in this system using canonical substrates? It is also notable that CLK2 undergoes autophosphorylation on Ser141, mutation changes its subnuclear localization (PMID 9852100). Does the K192R prevent this autophosphorylation? One potential explanation is that the autophosphorylated form of CLK2 is required to interact with p65 (whether due to TG003 or the K192R mutant), but does not reflect the requirement for its kinase activity.

Response:

We apologize for not using canonical substrates to directly verify the kinase-dead status of the CLK2^{K192R} mutant. However, we refer to the pivotal study by Joseph T. Rodgers et al. (DOI: 10.1016/j.cmet.2009.11.006), which demonstrated that CLK2^{K192R} lacks the ability to phosphorylate PGC-1 α in a γ^{32} P-ATP reaction, as illustrated in Supplementary Fig. 2D (presented below). This evidence firmly supports our assertion that the CLK2^{K192R} mutant is kinase-dead.

As the reviewer said “*It is also notable that CLK2 undergoes autophosphorylation on Ser141, mutation changes its subnuclear localization (PMID 9852100). Does the K192R prevent this autophosphorylation?*”. Addressing the reviewer’s insightful query, we highlight findings from the same study by Rodgers et al., showing γ^{32} P-ATP consumption when CLK2^{K192R} was co-incubated with CLK2^{WT} (Supplementary Fig. 2E and was presented below). This suggests that while CLK2^{K192R} cannot autophosphorylate, it can be phosphorylated by CLK2^{WT}. Furthermore, the localization pattern of CLK2^{K192R}, closely mirroring that of the CLK2^{S142A} mutant (DOI: 10.1074/jbc.273.51.34341), reinforces our conclusion regarding the kinase-dead status of CLK2^{K192R} and its inability to undergo autophosphorylation.

In our manuscript, the CLK2^{K192R} mutant was primarily utilized as a negative control, analogous to the use of TG003 as suggested by the reviewer. Moreover, the autophosphorylated form of CLK2 itself indicates the necessity of its kinase activity. We believe that regardless of the form of CLK2 - be it K192R, S142A, or treated with

TG003 - the K192R mutation indeed leads to the inactivation of CLK2 kinase activity, making our findings valid.

[REDACTED]

Rodgers JT, Haas W, Gygi SP, Puigserver P. Cdc2-like kinase 2 is an insulin-regulated suppressor of hepatic gluconeogenesis. *Cell Metab.* 2010;11(1):23-34. doi:10.1016/j.cmet.2009.11.006

- TG003 is not specific for CLK2, this drug also inhibits other CLKs. This is important since CLK1,2,4 each regulate other substrates including splicing factors (PMID 33846420). They need to demonstrate that this is not impacting other CLKs or that other CLK2 substrates aren't playing a role. Why not show that the impact of TG003 is only observed in CLK2+, but not CLK2-/- cells.

Response:

We are grateful for your insightful comments regarding the specificity of TG003. We acknowledge that TG003 is known to inhibit several members of the CLK family, including CLK1, CLK2, and CLK4. Recognizing the critical need to demonstrate the specificity of TG003's effects in the context of our study, we have undertaken additional experiments.

To unequivocally address the concerns raised, CLK2^{-/-} cells. Our results, now included as Supplementary Fig. 6m, show that TG003's presence or absence did not

influence the acetylation or phosphorylation levels of p65 in CLK2^{-/-} cells. This outcome strongly suggests that the impact of TG003 on p65 activation is specifically mediated through inhibition of CLK2, rather than other CLK family members.

-The guidelines and other publications that use the ADP-glo kinase assay report activity based on the change in luminescence with concentration of the protein/inhibitors present. Here, they apply it as a single timepoint/readout luminescent reading-they do not report kinase activity differences in CLK compared to the CLK2 K192R mutant over a range of protein/substrate concentrations. Given that CLK2 K192R is expressed at lower levels than CLK2 in Fig 4C- this decrease in luminescence might just represent a decrease in CLK2 mutant expression levels. They don't show assay results for FLAG-CLK2K192R in the absence of His-p65. What is the source of the samples examined (lysates or purified components)? The input proteins need to be shown to verify equal inputs. Minimally, a purified CLK2 kinase enzyme system is commercially available (Promega) and could be used with their purified p65 to verify p65 is a substrate for CLK2.

Response:

We are grateful for your valuable advice. Taking into account previous research and the suggestion of the technical adviser from Promega, we have decided on the specific concentration of CLK2 for the kinase assay.

It is so sorry for confusing you about the expression of CLK2 and CLK2^{K192R}, they were utilized with the same concentration which were examined by BCA Protein Quantification Kit. We deeply regret that we are unable to detect the sample for CLK2^{K192R} without His-p65 and did not show the examination of samples additionally.

To directly address your concerns, we procured the commercially available CLK2 kinase enzyme system from Promega. Given the unavailability of a commercial source for CLK2^{K192R}, we utilized the provided CLK2 enzyme and combined it with our own purified His-p65 for the kinase assay. The results clearly demonstrated a dose-dependent relationship between ATP consumption and the amount of His-p65 (0ng, 100 ng, 500 ng, 1000 ng, 2000 ng) with a constant 20 ng of CLK2 protein. These adjustments and additional experiments significantly bolster our confidence in the reported findings, establishing p65 as a substrate for CLK2. We believe these efforts adequately address your concerns and enhance the overall validity and robustness of our study.

-The rationale for pursuing S180 but not S316 does not seem logical since the alignments show similar patterns of conservation across species for both (Fig SB).

Response:

The question raised is a significant one, and as the statement in result section “The NF-κB luciferase reporter assay showed that p65^{S180D}, but not p65^{S316D}, completely blocked the wild-type p65-induced NF-κB luciferase reporter activation, suggesting that phosphorylation of p65 serine 180 is necessary for NF-κB deactivation (Supplementary Fig. 5d)”. Our research focuses on inhibiting NF-κB activation in innate immunity and inflammation, and serine 180 has shown to play a crucial role in this process. As a result, we highlight the importance of serine 180 in the deactivation of NF-κB signaling and chose to investigate the function of serine 180 in NF-κB signaling activation, while excluding serine 316.

Although Ser316 is also a potential phosphorylation site by CLK2, our experiments suggest that p65^{S316D} does not affect NF-κB signaling transactivation in response to viral and cytokine stimulation. This finding directs our focus towards Ser180 due to its significant impact on NF-κB signaling. Nonetheless, we acknowledge that Ser316 may play a role in other physiological or regulatory mechanisms. Recognizing this potential, we aim to investigate the function and implications of S316 phosphorylation in future studies, thus expanding our understanding of NF-κB signaling regulation and its broader physiological significance.

-The analysis of the p65 S180 mutant clearly shows that S180A is more active than S180D, but the stronger experiment is to show that both retain this profile with or without CLK2 expression, unlike WT p65 (Fig 5E).

Response:

Your concern is very important, and we have previously observed that S180A maintains its profile with or without CLK2. We conducted a similar experiment by overexpressing CLK2, and the results of the Luciferase assay showed that CLK2 overexpression does not affect the activation of NF-κB by p65^{S180A}, contrasting with the significant inhibition observed in the case of wild-type p65. Regarding the S180D mutant, which represents a phosphorylation-mimicking, deactivated form of p65, we chose not to include it in this assay. The decision was based on its intrinsic deactivated

state, which we anticipated would not provide additional insights into the impact of CLK2 overexpression on NF- κ B activation. The results are available and presented below, but are not included in the manuscript.

-There was a notable difference in the expression level of HA-S180D in Fig 5 due to a decrease in stability; that is the simple explanation for loss of NF- κ B activation in Fig 4 functional assays. Why aren't these assays shown in the presence and absence of CLK2 to demonstrate that only the WT p65 half-life/instability (Supp Fig 6G) is different between CLK2⁺ and CLK2^{-/-} cells (S180A and S180D would not show any dependence on CLK2 correct, correct)?

Response:

Thanks for your valuable suggestion. As what another reviewer said, “While MG132 experiments provide striking qualitative data demonstrating a role for phosphorylated ser180 in protein degradation, authors must calculate p65 wt, p65 S180A and p65 S180D half-life by performing a time course of cycloheximide (CHX) treatment, thus straightening their findings with solid quantitative data”. We had performed the experiment in HEK293T, HEK293T-p65^{S180A/KI}, and HEK293T-p65^{S180D/KI} cell lines then to further verify the stability of p65^{S180A} and p65^{S180D} following the reviewer #2's advice.

What you concerned are both reasonable and crucial. In accordance with your advice, we have performed the CHX assay in HEK293T-CLK2^{-/-} cells and the findings have been incorporated into Supplementary Fig. 6g. The results indicate that p65^{WT} in CLK2^{-/-} cells exhibits slightly greater stability than p65^{WT} in HEK293T cells, resembling the result of p65^{S180A/KI} closely. On the other hand, the consistently phosphorylated mutant p65^{S180D/KI} demonstrates notably inferior stability.

This data indicates that both p65^{WT} in CLK2^{-/-} cells and p65^{S180A/KI} exhibit comparable half-lives, suggesting that their stability is not influenced by CLK2-mediated phosphorylation. Accordingly, we infer that mutations at the Ser-180 site, namely S180A and S180D, exhibit no dependency on CLK2 for their stability or degradation profiles.

To reflect these insights, the result section has been revised as "and a cycloheximide (CHX) assay also revealed that $p65^{S180D}$ had a significantly shorter half-life than $p65^{WT}$. On the contrary, $p65^{S180A}$, which is similar to $p65^{WT}$ in $CLK2^{-/-}$ cells, showed a bit increased stability (Supplementary Fig. 6g)".

*-I remain in doubt about the findings using the Ser-180 p-p65 antibody (Fig 6F). The phospho- specific Ser-180 antibody that they had produced shows a non-specific band and evidently cross reacts with total p65, As controls, did they profile WT p65, S180A and S180D mutants to show this only recognizes the WT p65. Did they test p65-/- cells?*

**Response:**

We appreciate your concerns regarding the specificity of our phospho-specific Ser-180 p65 antibody. Although we did not profile S180A and S180D mutants with this antibody, we conducted specificity tests using $p65^{-/-}$ cells as a critical control. Supplementary Fig. 6k (which was presented below) demonstrates that the level of phos-p65 (Ser180) showed dynamic changes during SeV or TNF-α infection. Importantly, in $p65^{-/-}$ cells, where no p65 protein should be present, only a very weak band was observed. We interpret this minimal signal as non-specific binding, which is common in immunoblotting but does not detract from the overall specificity of the antibody towards its intended target under the conditions tested.

This control experiment supports the specificity of our phospho-Ser-180 p65 antibody, as significant reactivity is observed in the context of infections where Ser180 phosphorylation is expected to occur, while minimal background is detected in the absence of the p65 protein. We believe these findings adequately address the concern about antibody specificity and affirm the reliability of our results involving the detection of phospho-Ser180 p65.

-Also there is inconsistency between the CLK2^{+/+} vs -/- cells in Fig6D at basal (Ohrs) conditions. The p-p65 Ser180 is the same at Ohr in the SeV blot on left but higher in CLK2^{+/+} cells compared to CLK2^{-/-} cells at 0 hrs in the right TNF α blot.

Response:

We are grateful for your keen observation and it is exactly what you describe. We also feel confused. After reviewing multiple instances of repetitions, we believe that the p-p65 Ser180 level in CLK2^{+/+} cells should be higher than in CLK2^{-/-} cells at 0 hrs (The results of repetitions were presented below). This discrepancy may have been caused by a system error in a single experiment. Similar to the identification of the antibody in p65^{-/-} cells shown in the figure below, we can find that the blot at 0 hrs in HEK293T cells and blot in p65^{-/-} cells have a similar level of p-p65 Ser180. In conclusion, the specificity of p-p65 Ser180 has been confirmed, and the discrepancy may simply be an error that does not affect the conclusion we have reached.

2. I am still not convinced that CLK2 function is independent of I κ B α .

"Interestingly, there was an increased accumulation of Nfkb α in Clk2^{-/-} cells, which is consistent with the increase in Lys-310 acetylation and Ser-536 phosphorylation of p65. It indicates that CLK2 may be a basal checkpoint to limit constitutional activation of NF-KB signaling, even in the absence of activating stimuli."

-I remain unclear what they mean by 'early stage termination.' It seems that the impact of CLK2 on p65 and NFk β is most apparent in the context of induction than basal conditions. I κ B α is expressed and plays a role in basal and induced conditions over a 6 hr timecourse.

Response:

We apologize for any confusion caused by our previous explanation and we appreciate the opportunity to further clarify the role of CLK2 in the regulation of NF- κ B signaling and its independence from I κ B α -mediated termination.

Our focus is on the termination of transcriptional activation of p65. One well-known model involves I κ B α interacting with p65 in the cytoplasm, known as I κ B α -dependent termination. However, research has shown that p65 can still be retained in the cytoplasm even in the absence of I κ B α (doi: 10.1128/MCB.16.5.2341), suggesting the existence of other termination mechanisms independent of I κ B α . Our study reveals that CLK2 phosphorylates p65, regulating its post-translational modifications and reducing its DNA-binding activity and stability, ultimately leading to the termination of its transcriptional activation. Our proposed model is distinct from the I κ B α -dependent termination model, and thus we refer to it as I κ B α -independent termination.

According to the detailed explanation of the classic I κ B α -dependent termination model, I κ B α primarily inhibits p65 during basal conditions and in the later stage of activation. In basal conditions, I κ B α binds to p65 in the cytoplasm, preventing its nuclear translocation. Once the NF- κ B signaling is activated, I κ B α is mostly degraded, allowing p65 shuttles into the nucleus. Meanwhile, as one of the downstream genes of p65, I κ B α is resynthesized, mediated by p65 in the later stage, and binds with nuclear-p65 to bring it back to the cytoplasm (DOI: 10.1016/j.cell.2008.01.020). The defect of this model is that in the early stage of activation, there is no inhibitor to suppress or terminate the transcriptional activation of p65 when I κ B α has been mostly degraded but not yet been resynthesized. Therefore, the “early stage” here represents the period in which p65 shuttles into the nucleus and starts the progression of transcription in the absence of I κ B α . However, CLK2 can inhibit the activation of NF- κ B signaling once p65 shuttles into the nucleus, and the inhibition lasts until the termination of NF- κ B signaling. The novel termination mechanism by CLK2 includes the early stage of activation and is not dependent on I κ B α . Finally, this is referred to as “I κ B α -independent early stage termination”.

The change of p65 and Nfkbia is indeed more apparent in the context of induction than basal conditions. During activation, p65 is subject to much post-translational modifications, while Nfkbia undergoes mostly degradation and then re-synthesis.

About the statement “*I κ B α is expressed and plays a role in basal and induced conditions over a 6 hr time course*”. Actually, the Nfkbia we mentioned here is not being investigated for its function in basal and induced conditions. Instead, the expression level of Nfkbia serves as an indicator of the transcriptional activity mediated by p65, which is known to induce Nfkbia expression. It is noteworthy that the expression of Nfkbia in Clk2^{-/-} cells is more than that in Clk2^{+/+} cells, even under basal conditions (Fig. 2e and Supplementary Fig. 2f). The result indicates us that p65 displays heightened transcriptional activity in the absence of CLK2, no matter in basal or induced conditions.

-They state on lines 81-82 that 'we report an I κ B α -independent nuclear NF- κ B termination model at the transcriptional level through inhibitory phosphorylation by CLK2' Given that they have greater interactions of p65 with I κ B α in the presence of CLK2 (without or with TNF) in Supp Fig 61, doesn't this again indicate that I κ B α can

be playing a role in CLK2's inhibition of p65? Can they demonstrate that CLK2 suppresses p65 even in the absence of IκBα?

Response:

What a great question. As we answered previously, phosphorylation of p65 by CLK2 has been shown to inhibit the transcriptional activation of p65, and it was reflected by the post-translational modification levels of p65. This inhibition by CLK2 is independent of IκBα. Meanwhile, the interaction between IκBα and nuclear-p65 is regulated by the acetylation of p65. According to the work by Lin-feng Chen (DOI: 10.1093/emboj/cdf660), we can understand that acetylation of p65 at K221 and K218 can enhance the DNA binding of NF-κB and impair its association with IκBα.

However, it is important to differentiate between the effect of CLK2 on p65 activity and the role of IκBα. In our manuscript, we found that CLK2 inhibits several post-translational modifications of p65, which may also impair the acetylation levels of p65 at K221 and K218. Therefore, it makes sense that p65 has greater interactions with IκBα in the presence of CLK2. Moreover, phosphorylation by CLK2 and interaction with IκBα are two separate events, with a precedence relationship in time. The increased interaction between IκBα and p65 is one of the results caused by phosphorylation by CLK2. In a word, the inhibition of p65 by CLK2 is not affected by the absence or presence of IκBα, so it does not indicate that IκBα plays a role in CLK2's inhibition of p65.

During the activation of NF-κB signaling, IκBα is mostly degraded and re-synthesized. We can observe that active-p65 (phosphorylated and acetylated p65) can still be inhibited even when IκBα is mostly degraded in the presence of CLK2 (the 30 min time-course of LPS treatment in MLFs in Fig. 2e, and the 10 min time-course of TNF-α or IL-1β treatment in MLFs in Supplementary Fig. 2f). Therefore, we can demonstrate that CLK2 can still suppress p65 in the absence of IκBα.

-I don't understand their statement that IκBα functions in CLK2 KO cells- the data they cite just shows Nfkb1a is resynthesized by western. Do they introduce IκBα into CLK2^{-/-} cells and show it is still inhibitory?

Response:

Thank you for the opportunity to clarify this aspect of our study. It's important to understand that IκBα functions as both an inhibitor of NF-κB p65 and as a gene whose expression is induced by NF-κB activation. The results indicate that p65 displays enhanced transcriptional activity in the absence of Clk2, which result in the increased production of Nfkb1a. Actually, the expression level of Nfkb1a was observed as a downstream expressed protein in this context.

Regarding the question “Do they introduce IκBα into CLK2^{-/-} cells and show it is still inhibitory”, we need to explain that IκBα inhibits p65 by interacting with it, and this interaction is dependent on the acetylation level of p65 which is directly regulated by acetylases and deacetylases, such as p300/CBP or HDAC3 (doi: 10.1093/emboj/cdf660). In the absence of CLK2, we can still observe that p65 has

affinity to I κ B α , and still can interact with it. In general, I κ B α still exhibits its inhibitory function, although it is weakened in CLK2^{-/-} cells.

-They find that S180D (which does not activate NF- κ B and has a higher turnover) has a higher affinity for Nfkb1a/I κ B α (Supp Fig 6H), but S180A and WT in CLK2^{-/-} cells has a decreased interaction with CRM1 and I κ B α (Fig 6C & 6D). This data demonstrates that the phosphorylation on S180 induced by CLK2 leads to a higher affinity for I κ B α that would export and lock p65 out of nucleus; in turn this lead to the loss of NF- κ B post-translational modifications and functions in the nucleus. Therefore, the impact of CLK2 on p65 Ser180 phosphorylation may be upstream of I κ B α interactions, but ultimately is determined by I κ B α export. It is not independent of I κ B α - just not dependent on newly synthesized I κ B α in late stage termination. This does not fit with their conclusions and title.

Response:

We apologize for our previous explanation being unclear. As we mentioned before, I κ B α serves as a method for nuclear exportation, but p65 can also translocate to the cytoplasm independently of I κ B α (doi: 10.1128/MCB.16.5.2341). In our manuscript, the result shows that I κ B α can be used up in p65^{S180D/KI} cells (Supplementary Fig. 6e). Meanwhile, we observed that p65 can still shuttle from the nucleus to the cytoplasm in 1-3 hrs in p65^{S180D/KI} cells (Fig. 6b). Combine the results, they suggest that p65 can still be exported from the nucleus in absence of I κ B α . In summary, our conclusions reflect the broader understanding that CLK2 influences NF- κ B signaling through mechanisms that can both involve and bypass I κ B α . This nuanced view acknowledges the complexity of NF- κ B regulation, where multiple layers of control determine the nuclear-cytoplasmic dynamics of p65, including but not limited to its interactions with I κ B α .

-Lines 368-370 described showing p65 S180A and WT interactions with CRM1 and I κ B α in CLK2^{-/-} cells. Fig 6D only shows 293T cells and Fig.6i only shows WT p65 in both CLK2^{+/+} and CLK2^{-/-} cells.

Response:

We feel so sorry for confusing the reviewer. Fig. 6d displays the interaction between p65 and I κ B α or CRM1 in p65^{WT} and p65^{S180A/KI} cells (which is on right part of Fig, 6d). Supplementary Fig. 6 shows the interaction between p65 and I κ B α and CRM1 in CLK2^{+/+} and CLK2^{-/-} cells (both results are presented below). Therefore, the statement “Moreover, the p65^{S180A} and p65^{WT} in CLK2^{-/-} cells presented decreased interaction with CRM1 and I κ B α (Fig. 6d and Supplementary Fig. 6i)” accurately reflects the findings.

Minor comments.

Professional editing is required even with revisions. Numerous issues remain and some are newly introduced. 'plague' is still used where 'plaque' is appropriate.

Response:

We appreciate the reviewer's attention to detail and their advice on the necessity of professional editing to improve the manuscript's language quality. We have taken this feedback seriously and enlisted the services of SPRINGER NATURE Author Services (SNAS) to ensure the manuscript meets the highest standards of English language, grammar, punctuation, and phrasing. We have attached the editing certificate from SNAS as proof of this professional review and editing

[REDACTED]

We sincerely apologize for that we did not revised all the “plague” throughout the manuscript. Following your feedback, we have conducted a thorough review of the manuscript to correct this and any similar errors. We can now confirm that all instances have been corrected to “plaque”.

The IFA figures with brightfield added have become very small (Fig3, 5). The IFA for the paired flow cytometry data could be moved to supplemental or the other bar graphs in the figures could be shortened to make space.

Response:

We greatly appreciate your suggestion regarding the size of the IFA figures in Fig. 3 and 5. We recognize the importance of clearly presenting our results for effective communication of our findings. In response to your feedback, we have taken steps to enlarge the IFA figures in both Fig. 3 and 5, ensuring that the details are visible and the results can be easily interpreted.

These adjusted figures are now included in the revised manuscript. This modification aims to enhance the overall clarity and impact of our visual data presentation. Thank you for helping us improve the quality of our figures and, by extension, our manuscript.

I previously noted that the font kerning for the sequence alignment of Fig SB is off and that a monospace font is needed. The authors claim to have adjusted this- but the text is still not aligned properly.

Response:

So sorry for not revising the Fig. 5b correctly, we need to apologize for it. Due to the Arial font cannot be set as monospace, we have tired many different methods and it seems like not be better. To rectify this, we have now utilized the Courier New font, which supports monospace formatting, ensuring that the text is aligned properly across the sequence. This adjustment has been applied to Fig. 5b, and we are confident that it now meets the necessary standards for clear and accurate presentation.

	180
Human	VRDPSSGRPLRLPPVLSHP ¹⁸⁰ IFDNRAPNTAEL
Rat	VRDPSSGRPLRLTPVLSHP ¹⁸⁰ IFDNRAPNTAEL
Mouse	VRDPAGRPLLLTPVLSHP ¹⁸⁰ IFDNRAPNTAEL
Zebrafish	ITLSSGDLFPLEPVVSQPIYDNRAPNTAEL
Fruitfly	SEQKGRFTSPLPPVSEPIFDKKAMSDLVI
	316
Human	KKRRTYETFKSIMKSPFSGPTDRPPPPRR
Rat	KKRRTYETFKSIMKSPFNGPTEPRPPPPRR
Mouse	KKRRTYETFKSIMKSPFNGPTEPRPPPPRR
Zebrafish	KKRRTGMLHNLKLSIITGSSMSAERRPF
Fruitfly	RRKRQKTGGDPMHLLLQQQKQQLQNDHQD

There is a lack of discussion regarding its canonical substrates such as splicing factors vs this noncanonical functions of CLK2 to target p65. How does SeV, TNF α or other stimulations that activated CLK2 tie into its impacts on splicing? When CLK2 is activated by infection to target p65, what is the impact on canonical substrates?

Response:

Thanks for your reminding and we also think it is essential. The discussion has been updated and presented below.

“On the other hand, CLK2 can be regulated significantly by SeV, TNF- α or other stimulations. Activated CLK2 is usually considered to facilitate RNA alternative splicing processes involving SRSF1 and RBFOX2. When CLK2 is activated by infection to target p65, it may also promote the activation of SRSF1, which has been found that overexpression can inhibit the production of pro-inflammatory cytokines (doi: 10.1172/JCI127949). However, the promotion of RBFOX2, which can be regulated by CLK2, is considered to be positively associated with inflammation (doi: 10.1038/s42255-022-00681-y). In summary, these findings underscore the complex regulatory role of CLK2 in the inflammatory response, potentially exhibiting dual effects in different inflammatory diseases or virus infection”.

Typo in Supp Fig 4c legend line 1057: they state'The upper shifted band represents phosphorylated CLK2 protein.'This is actually p65, correct?

Response:

Sorry for making that mistake and the figure legend has been revised as “The upper shifted band represents the phosphorylated p65 protein”.

With regard to Fig 6D legend, they state that "The asterisk/upper band indicate the unspecific band that can be detected by the antibody and the explanation was added in the relative figure legends." This has not been corrected.

Response:

We feel so sorry that the statement of asterisk/upper band have not been corrected throughout the figure legends. Now, the statement “The upper band marked with asterisk indicates the unspecific band.” have been added in the figure legends of Fig. 6f, 6g and Supplementary Fig. 6k.

Reviewer #2 (Remarks to the Author):

Authors fixed all relevant issue raised, by performing new experiments as requested, and by modifying the manuscript, as suggested. I also encourage authors to make a further effort to improve manuscript written English (at pag 5 line 55, "rheumatoid arthritis", instead of "rheumatic arthritis"; at pag.6 line 89 "CLK2-deficient mice were observed to have higher levels", instead of "CLK2-deficient mice produce were observed to have higher levels"; at pag 22 line 444 "The transcription factor p65 is critical for cytokine-induced production of inflammation genes", instead of "The transcription factors p65 is critically for cytokine-induced production of inflammation genes"; at page 24 line 499" use of site-specific phosphorylation antibody", instead of "use of stie-specific phosphorylation antibody", and so on).

Response:

We are grateful for Reviewer #2's constructive feedback and the specific suggestions to enhance the manuscript's written English. In response, we have taken decisive action to address these concerns. The manuscript has undergone professional editing by SPRINGER NATURE Author Services (SNAS) to ensure accuracy in English language, grammar, punctuation, and phrasing. This comprehensive review aimed to correct the issues highlighted by the reviewer, including those mentioned and others throughout the document.

We have attached the editing certificate from SNAS as evidence of this professional language editing service. This step underscores our commitment to presenting our findings clearly and professionally, and we believe it significantly improves the manuscript's overall quality.

[REDACTED]

In addition, we appreciate the specific examples provided for language improvement and have made the following revisions to address these points:

The “rheumatoid arthritis” is instead of “rheumatic arthritis”. The “CLK2-deficient mice were observed to have higher levels” has been *revised as* “*Clk2-*

deficient mice had increased serum levels of inflammatory cytokines after viral infection and showed increased resistance to virus-induced death”. The “The transcription factors p65 is critically for cytokine-induced production of inflammation genes” has been modified as “The transcription factor p65 is critical for cytokine-induced production of inflammatory genes”. “the use of site-specific phosphorylated antibodies” is instead of “use of stie-specific phosphorylation antibody”.

Moreover, at pag. 21 line 440, I suggest to rephrase the sentence "...and TG003 may be developed as a cure of virus-infection" as "...and CLK2 inhibition may be exploited as a cure of virus-infection", due to the fact that TG003 inhibitor may not be suitable for further chemical optimization, to serve as an effective and safe antiviral.

Response:

We agree with the reviewer that TG003 used here may not be suitable. Under the modification of SNAS, the sentence has been modified as “...indicating that CLK2 may be a therapeutic target for inflammatory diseases and that inhibiting CLK2 may be a strategy to cure viral infection”.

REVIEWERS' COMMENTS

Reviewer #1 (Remarks to the Author):

The authors have adequately addressed my concerns and the concerns of reviewer 2 in the second revision.